# Application Layer-Based Denial-of-Service Attacks Detection against IoT-CoAP

**Sultan M. Almeghlef** [1,*], **Abdullah AL-Malaise AL-Ghamdi** [1,2], **Muhammad Sher Ramzan** [1]
**and Mahmoud Ragab** [3,4]

1   Information Systems Department, Faculty of Computing and Information Technology,
    King Abdulaziz University, Jeddah 21589, Saudi Arabia; aalmalaise@kau.edu.sa (A.A.-M.A.-G.);
    msramadan@kau.edu.sa (M.S.R.)
2   Information Systems Department, HECI School, Dar Al-Hekma University, Jeddah 34801, Saudi Arabia
3   Information Technology Department, Faculty of Computing and Information Technology,
    King Abdulaziz University, Jeddah 21589, Saudi Arabia; mragab@kau.edu.sa
4   Mathematics Department, Faculty of Science, Al-Azhar University, Naser City 11884, Cairo, Egypt
*   Correspondence: salmeghlef@stu.kau.edu.sa

**Abstract:** Internet of Things (IoT) is a massive network based on tiny devices connected internally and to the internet. Each connected device is uniquely identified in this network through a dedicated IP address and can share the information with other devices. In contrast to its alternatives, IoT consumes less power and resources; however, this makes its devices more vulnerable to different types of attacks as they cannot execute heavy security protocols. Moreover, traditionally used heavy protocols for web-based communication, such as the Hyper Text Transport Protocol (HTTP) are quite costly to be executed on IoT devices, and thus specially designed lightweight protocols, such as the Constrained Application Protocol (CoAP) are employed for this purpose. However, while the CoAP remains widely-used, it is also susceptible to attacks, such as the Distributed Denial-of-Service (DDoS) attack, which aims to overwhelm the resources of the target and make them unavailable to legitimate users. While protocols, such as the Datagram Transport Layer Security (DTLS) and Lightweight and the Secure Protocol for Wireless Sensor Network (LSPWSN) can help in securing CoAP against DDoS attacks, they also have their limitations. DTLS is not designed for constrained devices and is considered as a heavy protocol. LSPWSN, on the other hand, operates on the network layer, in contrast to CoAP which operates on the application layer. This paper presents a machine learning model, using the CIDAD dataset (created on 11 July 2022), that can detect the DDoS attacks against CoAP with an accuracy of 98%.

**Keywords:** denial-of-service; IoT attacks; CoAP security; application layer; DTLS

## 1. Introduction

IoT is a massive network that connects low power and low resource devices to the internet and enables them to communicate with each other without human intervention [1]. The number of IoT devices is growing and is expected to reach approximately 75 billion by the year 2025 [2]. As a result, it can be stated that IoT will lead the development of a smarter world in the upcoming decades in different fields, such as smart homes, smart industries, and smart healthcare [3]. IoT relies on different protocols to exchange information between the devices and the internet, and is composed of three layers, the perception layer which includes sensors that gather data from the environment, the network layer which receives data from sensors and processes them accordingly, and finally, the application layer which receives information from the network layer [4]. In terms of communication protocols, there are different protocols that operate on the application layer, including CoAP, Message Queuing Telemetry Transport (MQTT), and Advanced Message Queuing Protocol (AMQP). Nevertheless, CoAP is preferred over other protocols due to its lightness, its interoperability

with low power, and low resource devices, in addition to the fact that it can be secured using DTLS [5]. CoAP is based on the REST model and enables the resources to be addressed from the server and accessed by the clients with the help of the standard HTTP methods, such as (GET, PUT, POST, and DELETE). CoAP is preferred over other application protocols due to its simplicity for developers and for being lightweight in terms of power consumption, communication, mobility, and portability [6]. As mentioned above, CoAP is targeted heavily by DDoS attacks. This attack is carried out by a group of infected devices (zombies) that are controlled by an attacker. When instructed by the latter to launch an attack, the former overwhelms the victim with a high volume of requests, which results in consuming the victim's resources and making it unavailable to legitimate users. Therefore, securing CoAP against DDoS attacks is important. Employing the DTLS protocol is one way of securing CoAP; however, DTLS suffers from communication overhead since it sends and receives six messages for the handshake process, which results in consumption of the constrained device's resources. Moreover, DTLS is not designed for constrained devices [6]. Similarly, LSPWSN is used to secure CoAP messages, but this protocol operates on the network layer while CoAP operates on the application layer. Motivated by the assumption that it is easier to detect a DDoS attack at the victim's end and easier to prevent it at the attacker's end [7], this work aims to propose a method to detect DDoS attacks against CoAP in the application layer. This study uses CIDAD dataset, which contains DDoS attacks (interception, modification, and duplication of CoAP messages). The dataset has ~11,000 samples, of which only 288 are malware. As a result, we extend the dataset to 100,000 samples with ~50% for each category (benign and malware) using Generative Adversarial Networks (GANs). In addition, four different ML classification models, namely, Naïve Bayes, Random Forest, SVC, and Decision Tree are employed since these ML methods have shown impressive results in classifying IoT attacks [8]. The proposed model gains an accuracy of 98% with the Decision Tree algorithm, which outperforms other algorithms. The research questions include:

RQ1: Is it effective to secure CoAP in the application layer from DDoS attacks?
RQ2: What are the CoAP-level features that can be dedicated to secure CoAP in its perimeter?
RQ3: What is the best machine learning technique that can be performed well in detecting DDoS attacks against CoAP?

The research gap is to find a method to secure CoAP in the application layer while not relying on the lower layers to vet the CoAP message and deliver it to the application layer. The main aim of this research is to find a dataset containing different DDoS attacks and build a machine learning model that classifies these packets as DDoS or benign in the application layer. Therefore, the main contributions of this work are as follows:

1- Extending and balancing the CIDAD dataset using GANs.
2- Focusing on the CoAP level features to ensure the security of CoAP in its vicinity.
3- Build a machine learning model that can classify the benign against malware with an accuracy of 98% using the decision tree algorithm.

### 1.1. IoT Overview

The IoT enables the internet-connectivity of small devices allowing them to send and receive data with little to no intervention by the user [9]. According to the anticipation of some researchers, 75 billion IoT devices will exist in the communication technology environment by 2025 [2]. IoT consists of three layers (as depicted in Figure 1): The perception layer that represents the physical sensors and actuators, the network layer that enables device-to-device and device-to-cloud communication, and the application layer that delivers the services to other devices or humans [2]. IoT architecture can be extended to have two more layers, namely, MAC and adaptation layer resulting in the so-called five-layer architecture [10]. Since IoT devices were heterogeneous and needed to communicate with different types of other devices, the Institute of Electrical and Electronics Engineers (IEEE) and the Internet Engineering Task Force (IETF) developed standardized protocols

for enabling this communication. Figure 2 depicts the five-layer architecture of the IoT network and the protocols that operate on each layer.

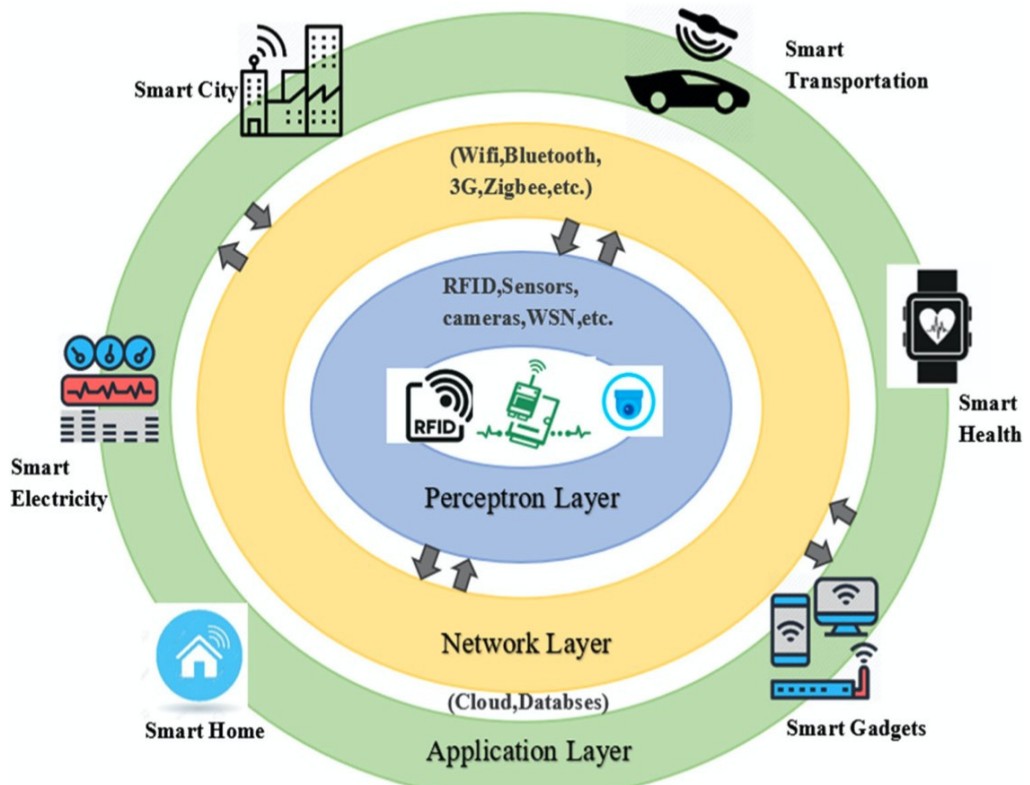

**Figure 1.** Layer architecture of IoT network. Adopted from [9].

| Application | • CoAP |
|---|---|
| Network | • RPL |
| Adaptation | • 6LoWPAN |
| MAC | • IEEE 802.15.4 |
| Physical | • IEEE 802.15.4 |

**Figure 2.** Layers and protocols of IoT network.

1.1.1. IoT Protocols

1- IEEE 802.15.4 Protocol

Both physical and MAC layers are ruled by the IEEE 802.15.4 protocol which emerged in 2003. The physical layer provides functionalities, such as transferring data, detecting channel energy, and indicating link quality [11], whereas the MAC layer associates network clients with the access points, acknowledges frame arrival, and validates frames.

2- The 6LoWPAN protocol

Low-Power Wireless Personal Area Network (6LoWPAN) emerged in 2007 to meet the demand for a low-energy IoT protocol [10]. It allows for a direct connection to the internet and defines encapsulation and header compression mechanisms. The 6LoWPAN is considered as a replacement for IPv6. This protocol supports addresses with different lengths, low bandwidth, and low costs.

3- Routing—RPL protocol

The IETF proposed the Routing Protocol Layer (RPL) for Low Power and Lossy Networks (LLNs), which provide IPv6 connectivity to LLNs [12]. This protocol is used

in multiple networking facilities, such as automated homes, automated industry, and automated buildings.

4- CoAP Protocol

The Constrained Application Protocol (CoAP) is a proper web transfer protocol for low resource devices and LLNs [13]. The IoT nodes often have 8-bit microcontrollers and limited RAM and ROM. The protocol supports Machine-to-Machine (M2M) applications, such as automated smart homes. CoAP emerged due to the demand for a generic web protocol that is suitable for all constrained devices.

### 1.1.2. CoAP Architecture

CoAP relies on the client/server model, such as the HTTP and uses the request/response model for exchanging messages. A CoAP request is similar to an HTTP request which asks for a resource on a server. Then, the resource is identified by an URI and a response code from the server is sent back with the representation of the resource. The CoAP architecture includes the message layer and the request/response layer as the main layers, see Figure 3. The former enables message delivery using the UDP protocol which supports optional reliability, and the latter removes outdated and duplicate messages using the request and response codes.

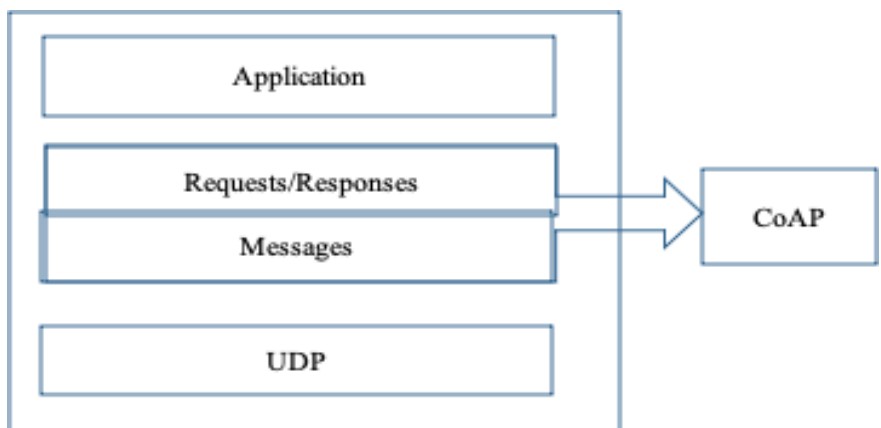

**Figure 3.** Layers of the CoAP protocol.

Messaging Model

Messaging model of CoAP is based on sending and receiving messages over UDP between two nodes. CoAP has a 4-byte header, options, and payload. Each message has a message ID for duplication check and a reliable connection if desired. Four types of messages that are supported by CoAP are as follows:

1- Confirmable Message (CON): In this mode, all messages are marked as confirmable messages (reliable mode). The message is resent using a default timeout until an ACK (acknowledgement) is received from the recipient with the same message ID as the sender. If the recipient fails to process the confirmable message, it will send an RST (reset) message rather than ACK to reset the communication. Figure 4 shows the confirmable mode between the client and the server.

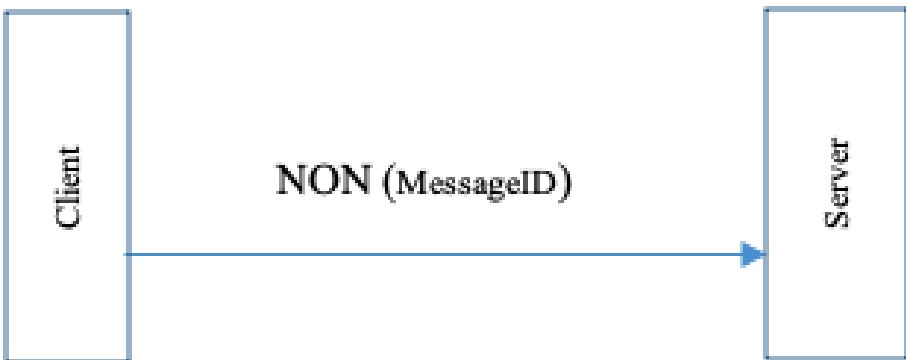

**Figure 4.** Confirmable message transmission.

2- Non-Confirmable Message (NON): If reliable delivery is not desired, the message can be sent as a non-confirmable message (unreliable mode). For duplication checking purposes, each message is assigned an ID even though its receival is not to be acknowledged. Figure 5 depicts the NON-message between the client and the server.

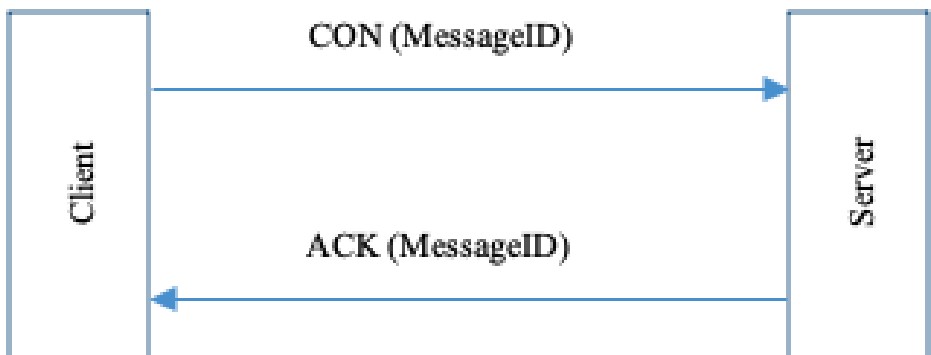

**Figure 5.** NON-Confirmable message transmission.

Request/Response Model

Request and response semantics which include method/response code are carried out by the CoAP message. The request and response code can include additional information, such as the URI and payload media type.

To check the identity of requests and responses, a "token" is used to connect responses with their corresponding requests. Conceptually, a token is different from a message ID. Following are the three types of messages used in the request/response model:

Piggyback message: A CON- or NON-message carries a request and if instantly enforced, the response is carried in the resulting ACK message. Figure 6 shows two examples of a basic GET request with a piggyback response, one for success and the other resulting in a 4.04 (Not Found) response. Therefore, the code [0x00] is the message ID.

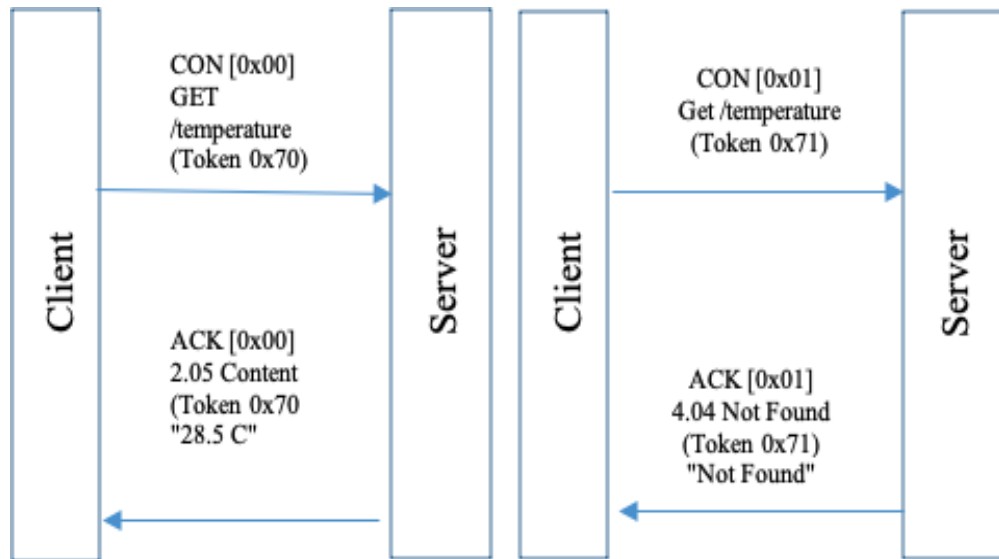

**Figure 6.** Two GET requests with piggyback responses.

Empty Message: If the server is not able to respond immediately to a request carried out on the CON-message, an empty acknowledgement is sent.

This empty acknowledgement response prevents the clients from re-transmitting the request. If the response is ready, the server sends it in a new confirmable message as depicted in Figure 7.

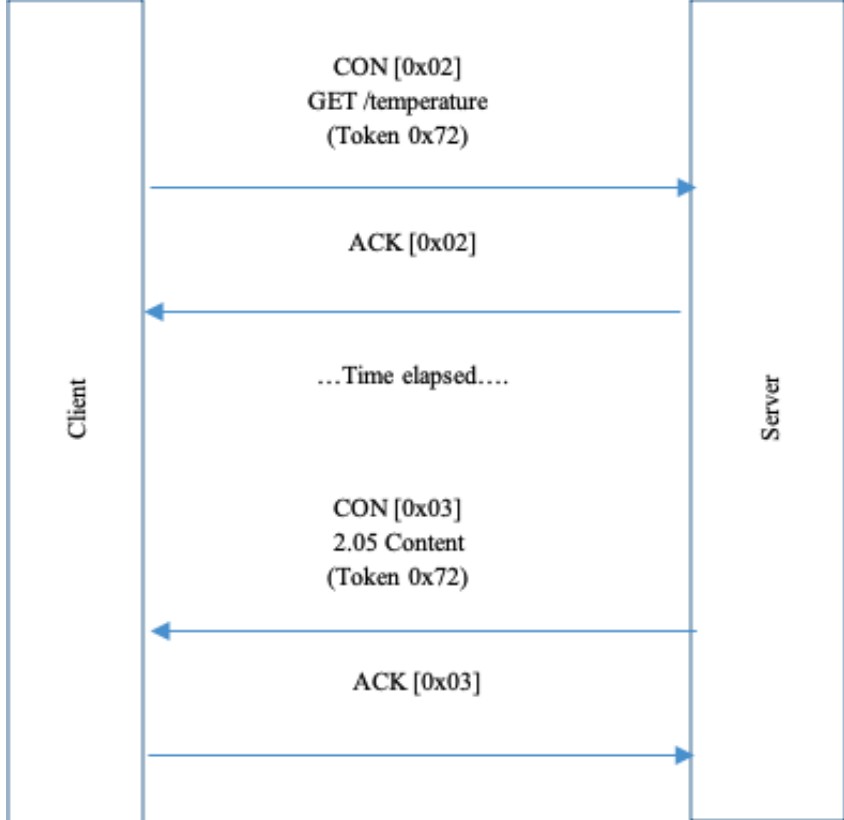

**Figure 7.** A GET request with separate responses.

Non-Confirmable Message: If a Non-Confirmable mode has been used to send a message, then a new Non-Confirmable or a Confirmable message is sent back as a response as illustrated in Figure 8.

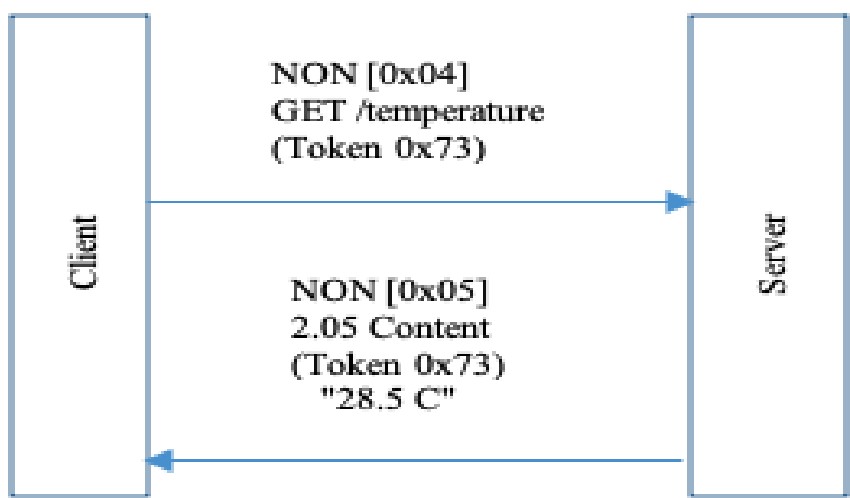

**Figure 8.** A Non-Confirmable message carries the request and response.

Message Format

CoAP uses UDP for sending and receiving messages and encodes its messages in a simple binary format. A fixed-size 4-byte format appears in the header of the message, then it is followed by a Token value of 0–8 bytes in length. After the Token value, CoAP Options in Type-Length-Value (TLV) format appear or a sequence of zeros for non-option is displayed. Finally, an optional payload appears that takes up the rest of the datagram. Figure 9 depicts the CoAP message format. Table 1 shows the header fields can be elaborated as follows:

(1) Version (V): Unsigned integer (2-bit) that represents the CoAP version number. This field takes (01 binary), and other values are reserved for future versions. If the message comes with unknown version numbers, it must be ignored.

(2) Type (T): Unsigned integer (2-bit) that represents 0 for Confirmable, 1 for Non-Confirmable, 2 for Acknowledgement, or 3 for reset as illustrated in the previous section.

(3) Token Length (TKL): Unsigned integer (4-bit) with a length of 0 to 8 bytes. Length of 9–15 bytes is specialized for message format errors.

(4) Code: Unsigned integer (8-bit) that is further divided into the most significant bits (3-bit) and the less significant bits (5-bit). It is represented as "c.dd" ("c" can be 0–7 as a digit for the 3-bit, and "dd" can be two digits in the range from 00 to 31 for the 5-bit). The most significant bits view 0 for a request, 2 for a successful response, 4 for a client error response, or 5 for a server error response. The other most significant bits are reserved. The code 0.00 represents an empty message as a special case.

(5) Message ID: Unsigned integer (16-bit) used for duplicate vetting purposes. It is also used to match Acknowledgement/Reset messages to messages of type Confirmable or Non-Confirmable, respectively.

(6) Token: It is used to correlate requests and responses and can range from 0 to 8 bytes, based on the length stated in the TKL field.

(7) Options: The value can be 0 by another option or by the payload.

(8) Payload: If it exists, it is prefixed by a (0xFF) marker as a benchmark for payload start. To calculate the length of the payload, it is counted from the end of the marker until the UDP datagram end.

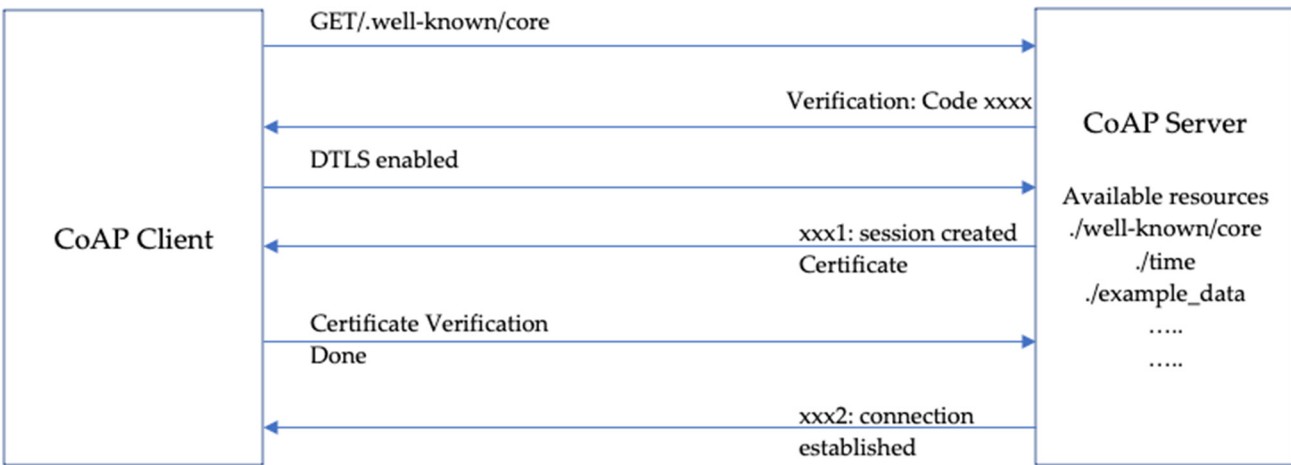

**Figure 9.** Cookie exchange technique by Maleh et al. [14].

**Table 1.** Message header of CoAP.

| Version (V) | Type (T) | Token Length (TKL) | Code | Message ID |
|---|---|---|---|---|
| | | Token (if any) | | |
| | | Options (if any) | | |
| | | Payload (if any) | | |

Method Definitions

Similar to HTTP, CoAP uses methods (GET, POST, PUT, and DELETE) to take any action on an URI resource. CoAP message follows RESTful architecture, which makes it appropriate for constrained devices as a lightweight protocol. A request that carries the fault method code results in a 4.05 (Unallowed method) piggyback response. We will briefly elaborate on each method.

GET

The GET method retrieves the information's representation that belongs to the resource identified by the request URI. If the GET method succeeds, a 2.05 (Content) is presented or a 2.03 (Valid) response code appears.

POST

The POST method's functionality is to process the representation retrieved by the request. The origin server performs the main functionality of the POST method depending on the target resource. This action results in the creation of a new resource or an update of the target resource. In the case of the creation of a new resource, the response should have a 2.01 (Created) code with the corresponding URI of the created source. However, if POST is processed but the creation of a new source on the server fails, then a 2.04 (Changed) response code is generated. If POST is processed and results in a deletion of a resource, the response should have a 2.02 (Deleted) response code.

PUT

The PUT method functionality is confined to creating or updating the resource identified by the request URI. The enclosed representation format is specified by the media type and content coding provided in the Content-Format option (if it is given). In the case of existing resources at the request URI, the enclosed representation is a modification copy, and a 2.04 (Changed) response code is issued. Otherwise, a new resource should be created by the server and aligned to that URI, and then, a 2.01 (Created) response code is issued. In the case of modifying or creating failure, the error response code is returned.

DELETE

The DELETE method requests to drop the resource that is identified by the request URI. If the deletion is a success, a 2.02 (Deleted) is issued. The same message is returned if the resource does not appear before the request.

CoAP URIs

The CoAP uses "coap" and "coaps" URI schemes to identify and locate resources. A CoAP server hierarchically organizes and governs these resources, and it waits to receive CoAP requests on a specified UDP port number. Therefore, the CoAP methods discussed above are used to access the resources on the CoAP server. Therefore, the "coap" and "coaps" URI schemes are similar to those of "HTTP" and "HTTPS", which are used with the HTTP protocol.

## 2. Related Technologies

It is a complicated task to secure IoT [1], as every layer in the IoT architecture, namely, (Application layer, Network layer, and Infrastructure layer) is susceptible to different kinds of attacks. Nevertheless, these attacks need to be identified and prevented by developing security models to ensure a secure IoT environment.

These attacks are caused by inherent vulnerabilities in the IoT environment. A vulnerability is a loophole that attackers exploit to penetrate the network. These vulnerabilities result in serious threats to the IoT network if ignored, and consequently, can lead to an attack. This work focuses on vulnerabilities that lead to DDoS attacks.

### 2.1. DDoS Attacks in IoT

A large amount of data flooding the network can result in the bandwidth being overwhelmed and the data server becoming inaccessible for serving new requests. This overflow of data is called a DDoS attack, in which legitimate users cannot access the server. While IoT is hailed as a revolution in technology, it can now also be considered as a bane since it attracts Botnets-based-DDoS attacks [9]. Botnets are defined as the process of infecting a massive number of IoT devices with malware to compromise these devices and bring them under the control of an attacker who can exploit them to launch an attack by ordering them to simultaneously send massive amounts of requests to the victim, and thereby consume its resources.

These infected IoT devices are called Bots. Normal IoT devices can be converted into Bots without the awareness or consent of their owners, and act as controlled slaves for the attacker or Master Bot Controller. Networks of these Bots are referred to as Botnets and are widely used for devious purposes these days and Master Controllers make profits by selling their "attack services". Following are four types of IoT botnets.

A- Mirai

Mirai is a Japanese word meaning the future. It is a malware that can be injected in over 500.000 non-secured IoT devices and can be used to overload a target server with massive traffic, with a flooding speed of 1 Tbps (Terabyte per second).

Mirai was designed to target Linux-running systems and, in 2016, it targeted the famous Dyn DNS (French hosting provider) service, causing major websites, such as Twitter, Amazon, and Netflix to go down. Subsequently, newer variants of Mirai emerged that were used to launch attacks on IoT devices.

B- Wirex

Wirex is a botnet used for launching DDoS attacks on multiple Content Delivery Networks (CDNs) and content providers. Wirex has infected an enormous number of Android devices using applications that appear to be benign applications but were malware. Consequently, Google banned some of these Wirex applications from its Play Store.

C- Reaper

This botnet is stronger than Mirai since the latter can only penetrate devices with default credentials. This malware created a large number of bots, including Cisco routers and other brands by brutally adding them to its botnet. Reaper can also exploit other vulnerabilities in IoT devices.

D- Torri

Torri is considered as a new botnet nowadays [15]. It can target most of today's recent computers, tablets, and smartphones due to its architectural design that has (64-bit), x86, etc.

### 2.2. CoAP Security Overview

This section introduces the DTLS binding for CoAP. Indeed, CoAP is equipped with the security demands, such as keying materials and an access control list during the provisioning phase or RawPublicKey mode. After the RawPublicKey mode finishes, the IoT device should be in one of the following security modes:

A- NoSec Mode: No security protocol is engaged (DTLS disabled). Alternatively, assuming that the lower-layer protocol will implement the security mechanism; therefore, the messages are transferred with no security.

B- Pre-sharedKey Mode: DTLS is enabled. This mode has a list of pre-shared keys and there is a list of nodes that is assumed to engage in the communication for every key. For instance, in a significant scenario, every node has its key if it engages in CoAP communication. In contrast, if a specific pre-shared key is shared with two entities or more, the entities are authenticated as a group by that key and would no longer be considered as specific peers.

C- RawPublicKey Mode: DTLS is enabled. This mode is used for device authentication. It provides each device with an asymmetric key that helps them in identifying and communicating with other devices without a certificate.

D- Certificate Mode: DTLS is enabled. In this mode, the asymmetric key paired with an X.509 certificate is reserved for a given device. The certificate is validated by a CA (Certification Authority).

### 2.2.1. Proposed Defense Mechanisms for Securing CoAP against DDoS Attacks

Some of the existing methods for securing the IoT network from DDoS attacks include the following. Saveetha et al. (2022) claimed that the intruder needs to discover the mapping of a network and it is hard to track all the scanning processes due to large network implementations [16]. Consequently, the authors developed an intrusion detection system (IDS) integrated with blockchain to detect the intrusions. Katib et al. (2023) claimed that blockchain has a significant role in IoT-based applications. Blockchain is used in many aspects, such as security and privacy in IoT-enabled deployment [17]. The authors proposed a hybrid Harris Hawks with sine cosine and a deep learning-based intrusion detection system to detect DDoS attacks against the IoT network. The BoT-IoT dataset was used to test their method and the model shows an impressive accuracy of 99.05%. However, these works are comprehensively dedicated for detecting attacks against IoT networks, while our focus is securing CoAP specifically from DDoS attacks since DDoS can target any layer on the IoT network architecture.

#### DTLS for CoAP Security

Some research proposed that DTLS can be used for CoAP security purposes. Maleh et al. (2016) stated that the Datagram Transport Layer (DTLS) handshake suffers from DDOS attacks based on IP spoofing [14]. To mitigate this threat, the DTLS handshake is expanded with a cookie exchange technique. According to this technique, the capability and threshold for receiving packets must be declared with the IP address to the server, thereupon the server reserves resources for new communication. Due to the high energy cost of this technique, the authors moved it to Proxy AC Server with no energy constraints.

Their method is depicted in Figure 9, and they claim that it reduces the resource occupancy of ROM by 23% compared to the standard DTLS. Haroon A. et al. (2017) proposed an enhancement to DTLS to make it resistant to DDoS attacks [18]. The authors claim that their method can reduce the overhead of handshaking time, packet size, and energy consumption compared to other works. The authors' system, named E-lithe, relies on Trusted Third Party (TTP) to reduce the overhead on the server-side. Compared to Lithe and other works, E-lithe outperforms others in terms of running time and reduced packet size. Later, this work was enhanced by Kajwadkar et al. (2018) who claimed that their work outperforms E-Lithe [19]. The authors' work essentially focuses on the comparison between the payload of benign and malicious packets, defining a threshold, which if exceeded, the source IP is blocked. They evaluated their work based on the malicious packet delivery ratio and legitimate packet drop ratio which outperforms the work carried out by Haroon et al.

SDN for CoAP Security

Alzahrani et al. (2020) implemented a software-defined networking scheme (depicted in Figure 10) to authorize the messages over the CoAP protocol [20]. The authors argue that the distributed approach, in which IoT devices employ powerful gateways attached to them, may not be sufficient, making the access control decisions accomplished by the controller render the security of CoAP messages more efficient and can help in avoiding DDoS attacks.

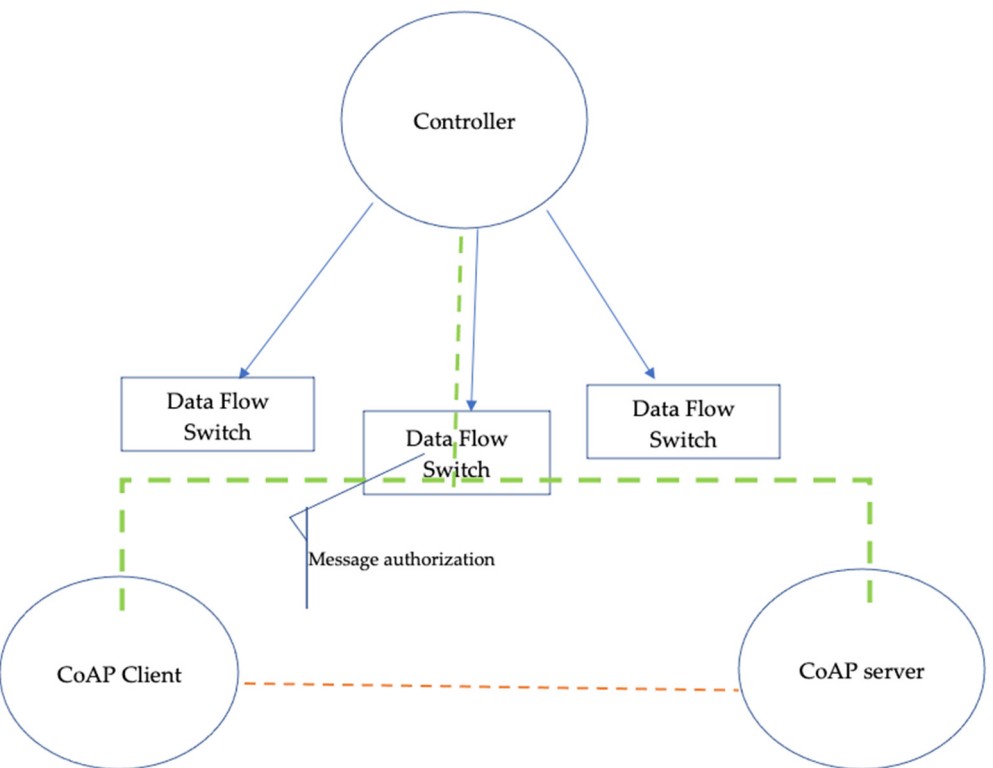

**Figure 10.** SDN-based schema for CoAP message authorization [20].

Machine Learning for CoAP Security

Machine Learning is also proposed for detecting and mitigating DDoS attacks against CoAP. Granjal et al. (2018) developed a framework that employs a threshold to mitigate DDoS attacks [21]. The authors define a limit for messages of CoAP, and after the limit is exceeded, extra messages are dropped. The authors enhanced their work and proposed an anomaly-based detection system to protect the 6LoWPAN and CoAP protocols from DDoS attacks [22], as depicted in Figure 11. SVM is used as an ML-Classifier and gains an accuracy of 93%. However, the system generates a high false-positive rate of around 20%. Doshi et al. (2018) developed a machine learning pipeline (depicted in Figure 12) that

is employed on middleboxes, such as routers or firewalls to detect IoT DDoS attacks [23]. They claim that IoT traffic is distinct from other traffic coming from other internet-connected devices since IoT traffic is repetitive and often communicates with a small finite of endpoints. After testing this method, it gains an accuracy of 99% using RF, KNN, and Neural Networks.

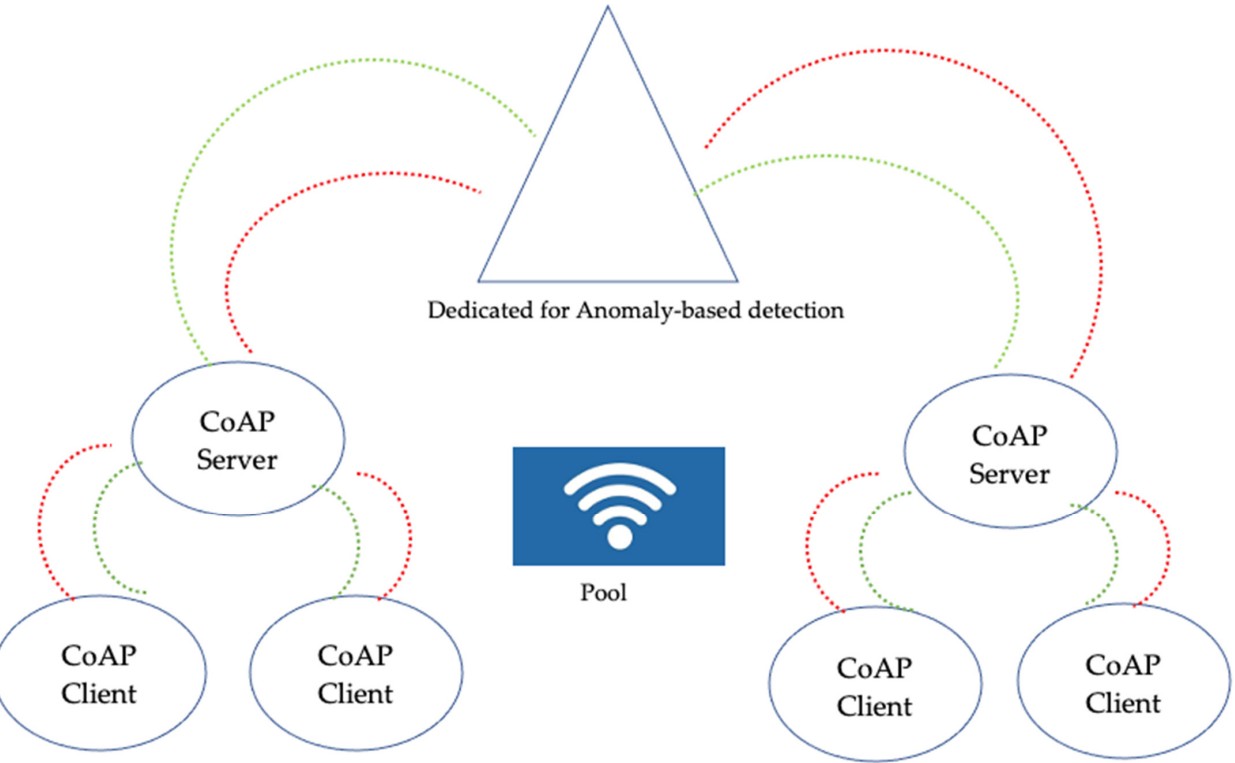

**Figure 11.** Anomaly-based detection framework for CoAP DDoS attacks by Granjal et al. [22].

Other Methods for CoAP Security

Other works propose different methods to cope with DDoS attacks against CoAP. Anirudh et al. (2017) [24] proposed a honeypot to lure the attacker and log his information for future verification or block purposes as depicted in Figure 13 and IoT-CoAP defense mechanisms in Table 2.

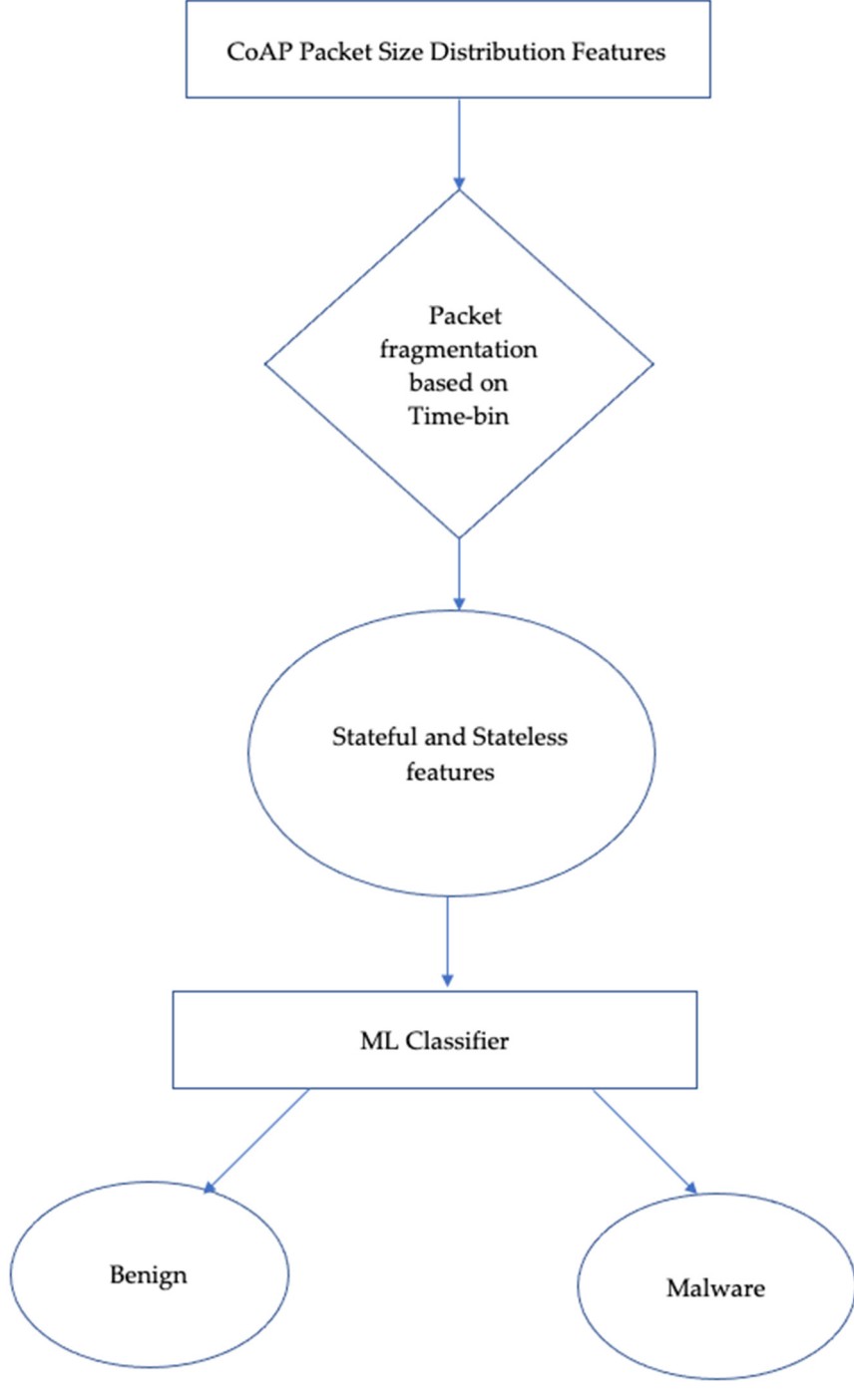

**Figure 12.** Machine learning-based pipeline to mitigate CoAP DDoS attacks by Doshi et al. [23].

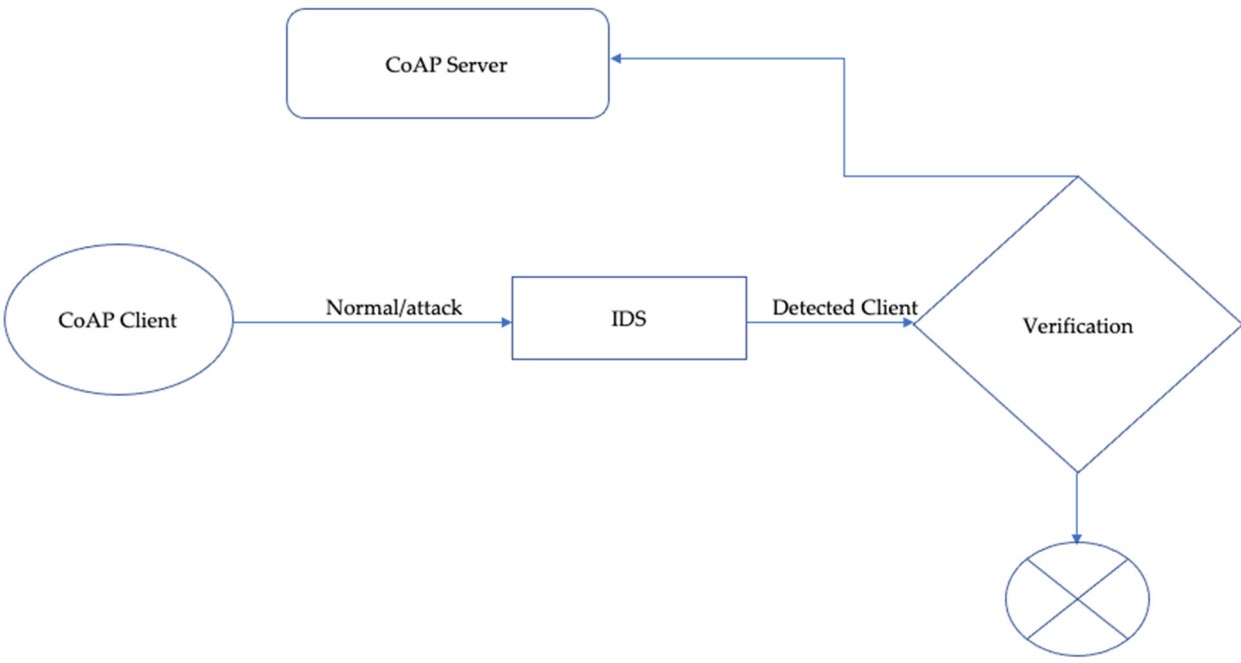

**Figure 13.** Honeypot-based detection method for mitigating CoAP DDoS attacks by Anirudh et al. [24].

**Table 2.** IoT-CoAP defense mechanisms.

| Research Objective | Methodology Used | Results | Limitation |
| --- | --- | --- | --- |
| Detect and mitigate DDoS attack against CoAP (2016) | DTLS handshake is extended with a cookie exchange technique to check the authentication of a message. | Low computation time by delegating all handshake to a third party. | The assumption of third party (Proxy AC server) is trustworthy all the time. |
| Secure DTLS for IoT (2017) | Trusted Third Party (TTP) is used to avoid DDoS attack and reduce overhead on the server side. | Energy consumption, reduced packet size, and reduced running time outperforms similar works. | DTLS is computationally heavy for IoT devices. |
| Deploy a honeypot to detect DDoS attack (2017) | Deploy a honeypot with two phases, first to log the anomalies and second to verify or block the client. | Sixty percent efficiency when a honeypot is implemented. | Anomaly-based detection may result in high false-positive rate. |
| Detect DDoS against IoT (2018) | Compare the payload of benign and malicious packet, define a threshold and if exceeded, the source IP is blocked. | Evaluate malicious packet delivery ratio and legitimate packet drop ratio. | Focuses on packet payload feature only. |
| Prevention framework from intrusion and DDoS attack (2018) | Relies on threshold by limiting the incoming request to a fixed number and if exceeded, the source request is blocked. | Fair energy and memory consumption when running the proposed system. | Susceptible to spoofing attack. |

**Table 2.** *Cont.*

| Research Objective | Methodology Used | Results | Limitation |
|---|---|---|---|
| Protect 6LoWPAN and CoAP from DDoS attack (2018) | Employ anomaly-based detection system to protect CoAP from DDoS attack. | SVM classifies the anomalies with accuracy of 93%. | High false-positive rate of 20%. |
| Detect IoT DDoS attack (2018) | Machine learning DDoS detection framework. | RF, KNN, Neural Networks gain approximately 99% accuracy. | Some IoT devices have different regular patterns but not distinct patterns. |
| Defend IoT against DDoS while maintaining benign traffic (2018) | Leverage the fast retransmit and flow control mechanism of TCP to retransmit benign packets at fastest rate and malicious packet at harmless rate. | Compared to D-WARD, FR-WARD performs better in retransmission, duration, and energy consumption. | Susceptible to other kinds of attacks. |
| Test CoAP MITM attack which results in spoofing, sniffing, and DDoS attack (2019) | Set up a client/server architecture to check whether the communication between the two devices using CoAP is secure. | Burp suite tool is used to intercept the communication between the client and the server. | Susceptible to sniffing attack. |
| Securing CoAP messages (2020) | SDN-based approach is developed to authorize the messages of CoAP. | Decrease overhead to the controller and CoAP responses become faster. | N/A |
| Design blockchain enabled IDS with deep learning (2022) | (IDS)-based deep learning integrated with blockchain to detect abnormal behavior in big networks. | The proposed model outperforms the conventional system in terms of accuracy. | N/A |
| Blockchain-Assisted Hybrid Harris Hawks Optimization-Based Deep DDoS Attack Detection in the IoT Environment (2023) | Hybrid Harris Hawks with sine cosine and a deep learning-based intrusion detection system to detect DDoS attacks against IoT network. | Obtain accuracy of 99.05% with BoT-IoT dataset. | N/A |

## 3. Materials and Methods

To secure CoAP in its vicinity, we focus on finding the dataset that contains CoAP-level packets only. As stated in Section 1.1, our target is to secure the CoAP in the application layer level. The CIDAD dataset is available on the Github.com [25], which is mainly generated to attack the CoAP in its vicinity. However, this dataset has fewer samples of the attacks and ~10,000 samples of the benign packets. Machine learning algorithms are always greedy for decent samples of data that can be learned. In the next section, we elaborate on how to extend the dataset to gain ~100,000 samples of DoS and benign packets.

### 3.1. Dataset Collection

The CIDAD dataset is targeting the CoAP with three different DDoS attacks: Duplication, interception, and modification of the CoAP message with a total of 288 malware samples. Interception means intercepting stochastically sent packets before reaching the destination, whereas duplication is changing the content of the CoAP message, and modification is increasing the number of tokens. The rest of the ~10,000 packets are benign packets. This poses an imbalance in the dataset since only 0.02% of the dataset is malware. Therefore, we extended the dataset to 100,000 samples, of which ~50% are benign and ~50% are malware. We used the Generative Adversarial Network (GAN) to extend the 288 malware samples to ~50,000 samples. To perform this, we used the Google Colab platform since the generated data needed a high-performance machine to be manipulated. The time to generate the fake data from the malware was higher, which took around 1

h since we had only 288 samples and we aimed to obtain ~50,000. However, the benign sample was adequate to generate the fake samples which took around 10 min. The generated samples were then compared with the original data to ensure the similarity between them by training each sample and calculating the Root Mean Square Error (RMSE). Our finding is surprising in that the RMSE is ~0.005 for original data, whereas for the generated data, it is ~0.09. This indicates a coherent similarity between the original malware and the generated ones. On the other hand, for the benign samples, we repeated the process and found that the RMSE for both the original samples and the generated ones is ~0.03. Figure 14 illustrates the general structure for GANs and Figure 15 shows the distribution of the collected dataset.

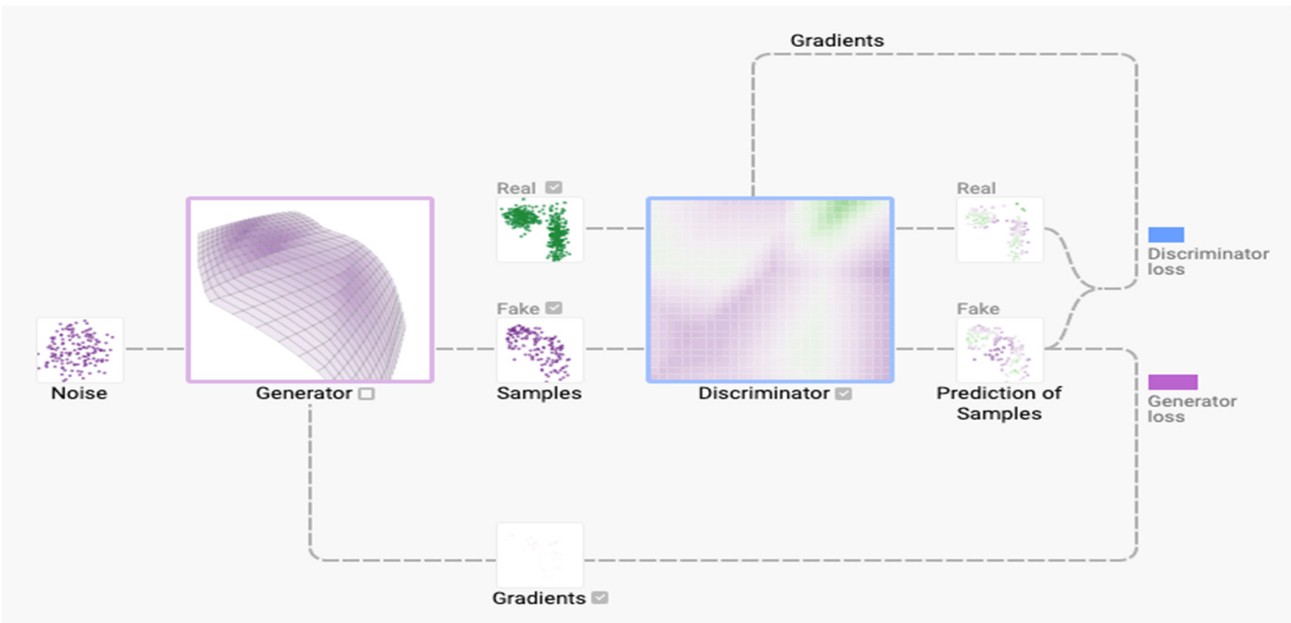

**Figure 14.** GAN architecture [26].

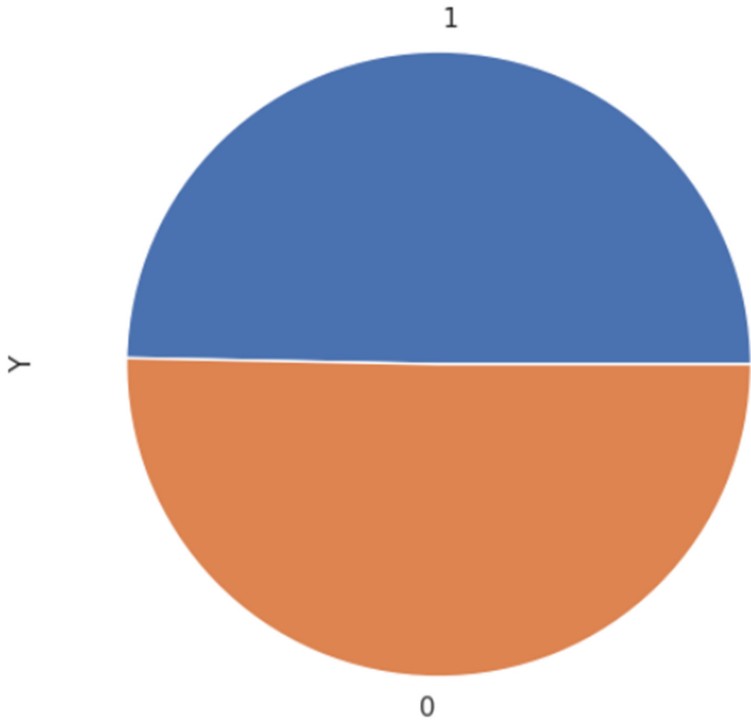

**Figure 15.** Dataset distribution (0 for benign, 1 for malware).

### 3.2. Feature Extraction

CoAP has a total of 86 features that can be extracted from the pcap file as shown in Table 3. However, most of them have a large number of missing values due to the optionality in the communication and since GANs cannot generate fake data for empty values of the features. Only 10 features have fewer missing values (less than 10% of the column data) in the dataset. Therefore, we focused only on these features as depicted in Figure 16. Then, we used the Pearson Correlation method formula (1) to achieve the relevant features to the labels (benign and malware).

$$r_{xy} = \sum_{i=1}^{n} \frac{(x_i - \overline{x})(y_i - \overline{y})}{\sqrt{\sum_{i=1}^{n}(x_i - \overline{x})^2}\sqrt{\sum_{i=1}^{n}(y_i - \overline{y})^2}} \tag{1}$$

where $r_{xy}$ is the Pearson correlation between two features $x$ and $y$, $n$ represents the total number of samples, $x_i$ and $y_i$ are the individual sample points indexed with $i$, $\overline{x}$ and $\overline{y}$ are the sample mean.

**Table 3.** Features associated with CoAP.

| Field Name | Description | Type | Versions |
|---|---|---|---|
| coap.block | Block | Frame number | 3.4.0 to 4.0.5 |
| coap.block.count | Block count | Unsigned integer (32 bits) | 3.4.0 to 4.0.5 |
| coap.block.error | Block defragmentation error | Frame number | 3.4.0 to 4.0.5 |
| coap.block.multiple_tails | Block has multiple tails | Boolean | 3.4.0 to 4.0.5 |
| coap.block.overlap | Block overlap | Boolean | 3.4.0 to 4.0.5 |
| coap.block.overlap.conflicts | Block overlapping with conflicting data | Boolean | 3.4.0 to 4.0.5 |
| coap.block.reassembled.in | Reassembled in | Frame number | 3.4.0 to 4.0.5 |
| coap.block.reassembled.length | Reassembled block length | Unsigned integer (32 bits) | 3.4.0 to 4.0.5 |
| coap.block.too_long | Block too long | Boolean | 3.4.0 to 4.0.5 |
| coap.block_length | Block Length | Unsigned integer (32 bits) | 3.4.0 to 4.0.5 |
| coap.block_payload | Block Payload | Byte sequence | 3.4.0 to 4.0.5 |
| coap.blocks | Blocks | Label | 3.4.0 to 4.0.5 |
| coap.code | Code | Unsigned integer (8 bits) | 1.6.0 to 4.0.5 |
| coap.invalid_option_number | Invalid Option Number | Label | 1.12.0 to 4.0.5 |
| coap.invalid_option_range | Invalid Option Range | Label | 1.12.0 to 4.0.5 |
| coap.length | Length | Unsigned integer (32 bits) | 3.2.0 to 4.0.5 |
| coap.mid | Message ID | Unsigned integer (16 bits) | 1.12.0 to 4.0.5 |
| coap.ocount | Opt Count | Unsigned integer (8 bits) | 1.10.0 to 1.10.14 |
| coap.opt.accept | Accept | Character string | 1.8.0 to 4.0.5 |
| coap.opt.block_mflag | More Flag | Unsigned integer (8 bits) | 1.6.0 to 4.0.5 |
| coap.opt.block_number | Block Number | Unsigned integer (32 bits) | 1.6.0 to 4.0.5 |
| coap.opt.block_size | Encoded Block Size | Unsigned integer (8 bits) | 1.6.0 to 4.0.5 |
| coap.opt.ctype | Content-type | Character string | 1.6.0 to 4.0.5 |
| coap.opt.delta | Opt Delta | Unsigned integer (8 bits) | 1.6.0 to 4.0.5 |
| coap.opt.delta_ext | Opt Delta extended | Unsigned integer (16 bits) | 1.12.0 to 4.0.5 |
| coap.opt.desc | Opt Desc | Character string | 1.10.0 to 4.0.5 |

**Table 3.** *Cont.*

| Field Name | Description | Type | Versions |
|---|---|---|---|
| coap.opt.end_marker | End of options marker | Unsigned integer (8 bits) | 1.12.0 to 4.0.5 |
| coap.opt.etag | Etag | Byte sequence | 1.6.0 to 4.0.5 |
| coap.opt.hop_limit | Hop Limit | Unsigned integer (8 bits) | 3.4.0 to 4.0.5 |
| coap.opt.if_match | If-Match | Byte sequence | 1.8.0 to 4.0.5 |
| coap.opt.if_none_match | If-None-Match | Byte sequence | 1.8.0 to 1.8.15 |
| coap.opt.jump | Opt Jump | Unsigned integer (8 bits) | 1.10.0 to 1.10.14 |
| coap.opt.length | Opt Length | Unsigned integer (8 bits) | 1.6.0 to 4.0.5 |
| coap.opt.length_ext | Opt Length extended | Unsigned integer (16 bits) | 1.12.0 to 4.0.5 |
| coap.opt.location | Location | Character string | 1.6.0 to 1.6.16 |
| coap.opt.location_path | Location-Path | Character string | 1.8.0 to 4.0.5 |
| coap.opt.location_query | Location-Query | Character string | 1.8.0 to 4.0.5 |
| coap.opt.max_age | Max-age | Unsigned integer (32 bits) | 1.6.0 to 4.0.5 |
| coap.opt.name | Opt Name | Character string | 1.10.0 to 4.0.5 |
| coap.opt.object_security_expand | Expanded Flag Byte | Boolean | 2.6.0 to 3.2.18 |
| coap.opt.object_security_kid | Key ID | Byte sequence | 2.6.0 to 4.0.5 |
| coap.opt.object_security_kid_context | Key ID Context | Byte sequence | 2.6.0 to 4.0.5 |
| coap.opt.object_security_kid_context_len | Key ID Context Length | Unsigned integer (8 bits) | 2.6.0 to 4.0.5 |
| coap.opt.object_security_kid_context_present | Key ID Context Present | Boolean | 2.6.0 to 4.0.5 |
| coap.opt.object_security_kid_present | Key ID Present | Boolean | 3.0.0 to 4.0.5 |
| coap.opt.object_security_non_compressed | Non-compressed COSE message | Boolean | 2.6.0 to 3.2.18 |
| coap.opt.object_security_piv | Partial IV | Byte sequence | 2.6.0 to 4.0.5 |
| coap.opt.object_security_piv_len | Partial IV Length | Unsigned integer (8 bits) | 2.6.0 to 4.0.5 |
| coap.opt.object_security_reserved | Reserved | Boolean | 3.4.0 to 4.0.5 |
| coap.opt.object_security_signature | Signature Present | Boolean | 2.6.0 to 3.2.18 |
| coap.opt.observe | Observe | Unsigned integer (32 bits) | 2.0.0 to 4.0.5 |
| coap.opt.payload_desc | Payload Desc | Character string | 1.10.0 to 2.2.17 |
| coap.opt.proxy_scheme | Proxy-Scheme | Character string | 2.0.0 to 4.0.5 |
| coap.opt.proxy_uri | Proxy-Uri | Character string | 1.8.0 to 4.0.5 |
| coap.opt.size1 | Size1 | Unsigned integer (32 bits) | 2.0.0 to 4.0.5 |
| coap.opt.subscr_lifetime | Lifetime | Unsigned integer (32 bits) | 1.6.0 to 1.12.13 |
| coap.opt.token | Token | Character string | 1.6.0 to 1.10.14 |
| coap.opt.unknown | Unknown | Byte sequence | 1.10.0 to 4.0.5 |
| coap.opt.uri_auth | Uri-Authority | Character string | 1.6.0 to 1.6.16 |
| coap.opt.uri_host | Uri-Host | Character string | 1.8.0 to 4.0.5 |
| coap.opt.uri_path | Uri-Path | Character string | 1.6.0 to 4.0.5 |
| coap.opt.uri_path_recon | Uri-Path | Character string | 2.4.0 to 4.0.5 |
| coap.opt.uri_port | Uri-Port | Unsigned integer (16 bits) | 1.8.0 to 4.0.5 |

**Table 3.** *Cont.*

| Field Name | Description | Type | Versions |
| --- | --- | --- | --- |
| coap.opt.uri_query | Uri-Query | Character string | 1.6.0 to 4.0.5 |
| coap.optcount | Option Count | Unsigned integer (8 bits) | 1.6.0 to 1.8.15 |
| coap.option_length_bad | Option length bad | Label | 1.12.0 to 4.0.5 |
| coap.option_object_security_bad | Invalid Object-Security Option Format | Label | 2.6.0 to 3.2.2 |
| coap.option_oscore_bad | Invalid OSCORE Option Format | Label | 3.2.3 to 4.0.5 |
| coap.oscore_kid | OSCORE Key ID | Byte sequence | 2.6.0 to 4.0.5 |
| coap.oscore_kid_context | OSCORE Key ID Context | Byte sequence | 2.6.0 to 4.0.5 |
| coap.oscore_piv | OSCORE Partial IV | Byte sequence | 2.6.0 to 4.0.5 |
| coap.payload | Payload | Character string | 1.12.0 to 4.0.5 |
| coap.payload_desc | Payload Desc | Character string | 2.4.0 to 4.0.5 |
| coap.payload_length | Payload Length | Unsigned integer (32 bits) | 2.4.0 to 4.0.5 |
| coap.request_first_in | Retransmission of request in | Frame number | 3.2.0 to 4.0.5 |
| coap.response_first_in | Retransmission of response in | Frame number | 3.2.0 to 4.0.5 |
| coap.response_in | Response In | Frame number | 2.2.0 to 4.0.5 |
| coap.response_time | Response Time | Time offset | 2.2.0 to 4.0.5 |
| coap.response_to | Request In | Frame number | 2.2.0 to 4.0.5 |
| coap.retransmitted | Retransmitted | Label | 3.2.0 to 4.0.5 |
| coap.tid | Transaction ID | Unsigned integer (16 bits) | 1.6.0 to 1.10.14 |
| coap.token | Token | Byte sequence | 1.12.0 to 4.0.5 |
| coap.token_len | Token Length | Unsigned integer (8 bits) | 1.12.0 to 4.0.5 |
| coap.type | Type | Unsigned integer (8 bits) | 1.6.0 to 4.0.5 |
| coap.unknown_option_number | Unknown Option Number | Label | 3.2.5 to 4.0.5 |
| coap.version | Version | Unsigned integer (8 bits) | 1.6.0 to 4.0.5 |

We focused only on the features that have $\pm 0.30$ correlation to the label. Figure 16 shows the correlation between the features and the label.

The Pearson Correlation for all the features is less than the target value $\pm 0.30$. Therefore, we used other methods which represent the statistical methods: Lasso Regression and One-Way ANOVA test. Lasso Regression is a regression analysis that deepens the accuracy and interpretability by performing regularization alongside the variable selection as shown in Formula (2). On the other hand, one-way ANOVA checks the significant independence of two or more samples, where the *p*-value decides a rejection for the null hypothesis of samples equality if the score is less than 0.05 as shown in Formula (3):

$$\sum_{i=1}^{tM}(y_i - \hat{y}_i)^2 = \sum_{i=1}^{M}\left(y_i - \sum_{j=0}^{p} w_j \times x_{ij}\right)^2 + \lambda \sum_{j=0}^{p}|w_j| \tag{2}$$

where *M* is the total number of samples, *P* is the feature, and *w* is the slope.

$$F = \frac{MSB}{MSW} \tag{3}$$

where *F* represents the ANOVA coefficient, *MSB* is the mean sum of squares between the samples, and *MSW* is the mean sum of squares within the samples.

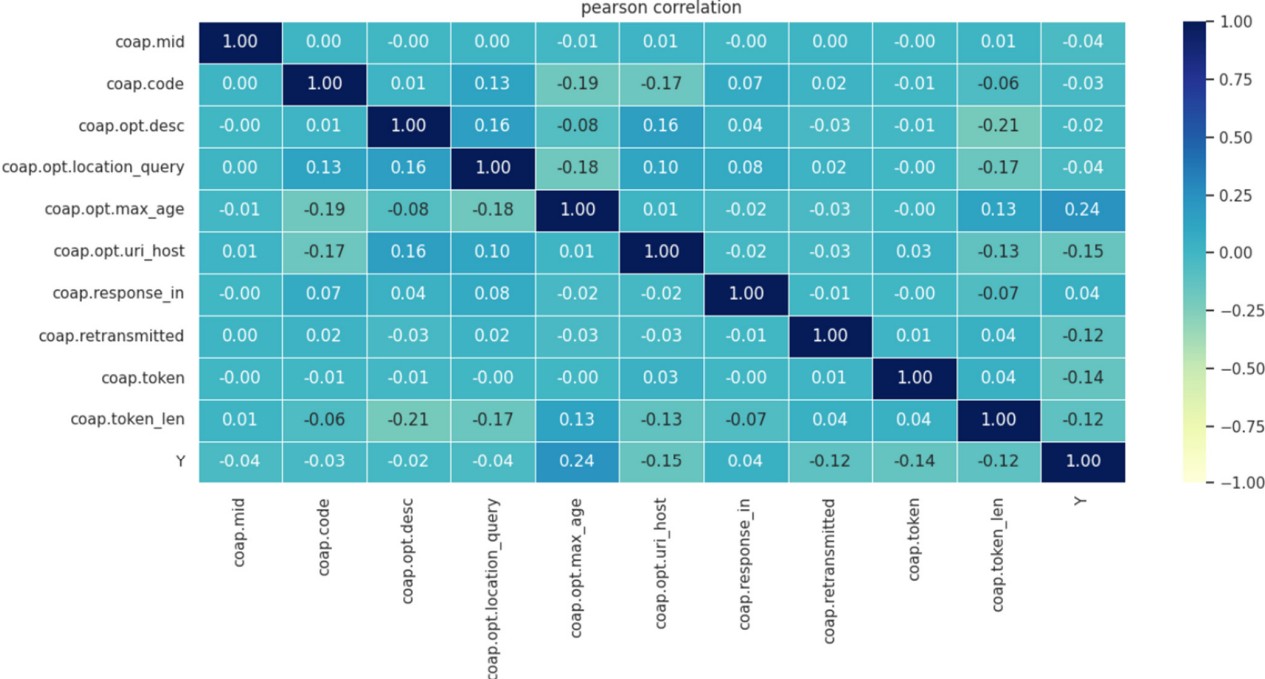

**Figure 16.** The Pearson Correlation between the features and the label.

We assume that any feature that is recommended by the two methods (ANOVA and Lasso) is correlated to the label and will be used for the training phase of the model. After calculating the Lasso and ANOVA methods, we found a total of six features that are recommended by both methods as depicted in Figure 17.

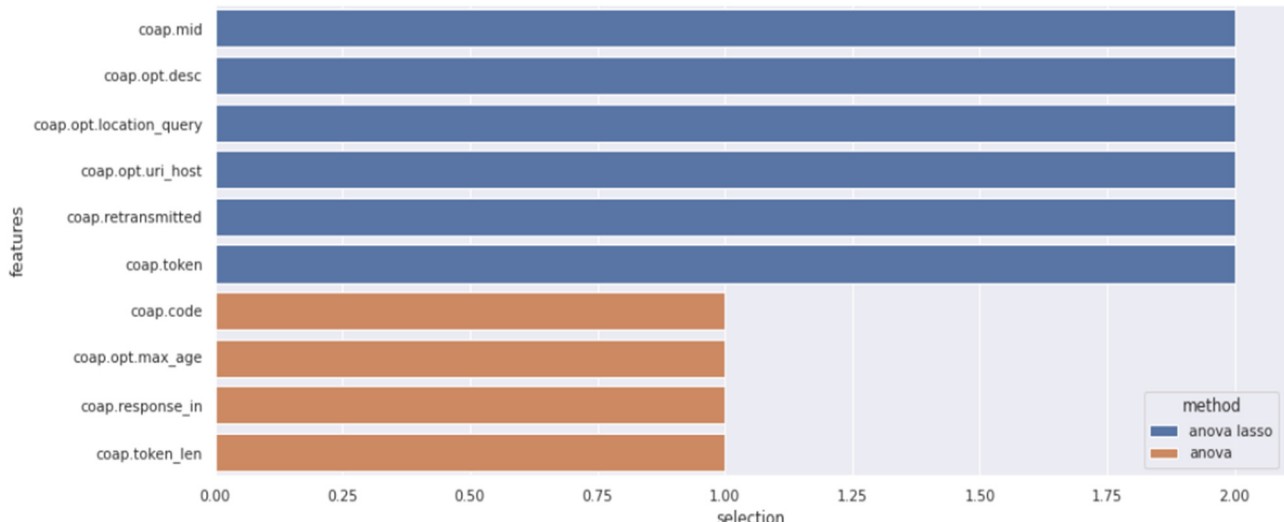

**Figure 17.** Lasso regression and ANOVA analysis for CoAP features.

The recommended features are coap.mid, coap.opt.disc, coap.opt.location_query, coap.opt.uri_host, coap.retransmitted, and coap.token. Table 4 shows the description for each feature. Only the features that are recommended by ANOVA are ignored.

**Table 4.** CoAP features description.

| Feature | Description | Type |
| --- | --- | --- |
| coap.mid | Message ID | Unsigned integer (2 bytes) |
| coap.opt.desc | Opt Desc | Character string |
| coap.opt.location_query | Location-Query | Character string |
| coap.opt.uri_host | Uri-Host | Character string |
| coap.retransmitted | Retransmitted | Label |
| coap.code | Code | Unsigned integer (1 byte) |

*3.3. Model Training*

We chose to test four machine learning classifiers (LinearSVC, Naïve Byes, Random Forest, and Decision Tree). We split the dataset into 70% for training and 30% for testing. The model is depicted in Figure 18.

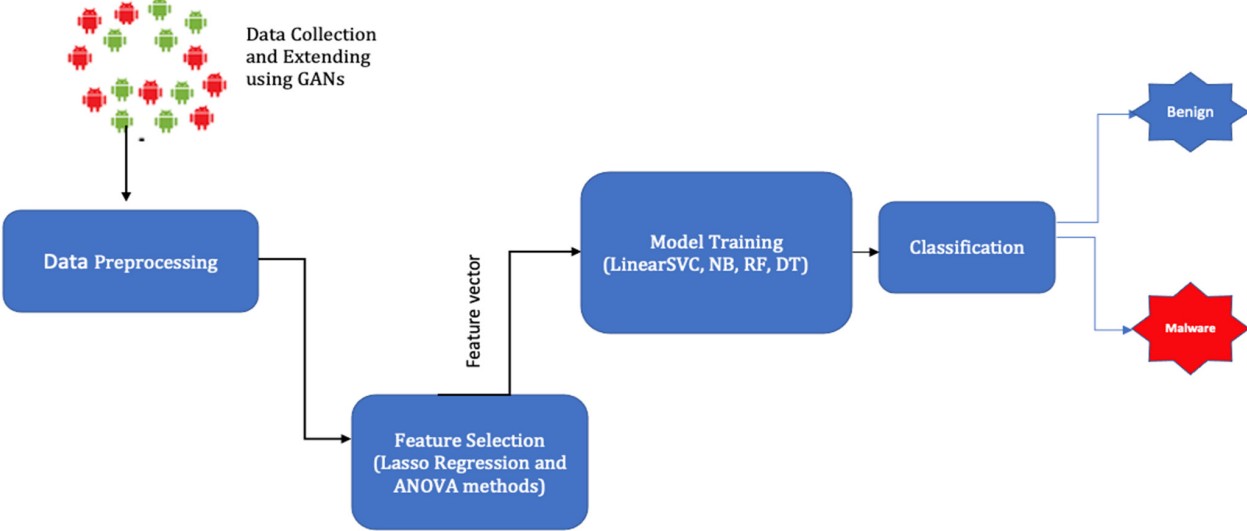

**Figure 18.** Proposed model.

We used Google Colab platform [16] to perform the experiment and evaluated the performance for each model. Since our data were ~100,000 samples; therefore, processing these data and building a model required a high capability computing platform, such as the Google Colab platform. The steps used to analyze the performance of the model are as follows:

1- Environment Setup: In this step, we imported the data after cleaning and normalizing it, as previously discussed in Sections 3.1 and 3.2 .
2- Analyzing the data: We selected only the features that are relevant to the label by calculating the Lasso regression and ANOVA analysis, which resulted in the selection of only 6 features from a total of 86 CoAP message features.
3- Data Preprocessing: In this phase, we handled the missing values by calculating the average of the column, and then filled the missing values with the average. Some of the features contained large numbers in order to solve this issue and keep the model calculations simple. Additionally, we normalized these values using the max-min technique to retain the values between 0 and 1.
4- Model design: For each classifier, we ran the experiment and validated the performance for each model using the cross-validation technique to avoid overfitting and underfitting. We set up the fold to 5 and calculated the accuracy for each round, and

then compared it to the model performance. Neither overfitting nor underfitting was observed in our experiment.

5- Evaluating the performance: The performance for each model was calculated in terms of accuracy, precision, recall, and F1-score. The following section elaborates on the results of the model performance in detail.

## 4. Results

### 4.1. LinearSVC

Linear Support Vector Classifier (SVC) is a type of Support Vector Machine (SVM). SVM is one of the machine learning algorithms that is used for supervised learning (labeled data), such as detecting fraud, outliers, and even regression problems. It simply draws a line between two categories by putting similar data points in one class and the remaining in the other. The overall result may contain several lines for classifying data points. Compared to K-Nearest Neighbor algorithm, SVC classifies the data point while mitigating the close data point to the line. This can be achieved by what is called decision boundary. Therefore, SVC relies on the features to find the decision boundary and the line can be substituted by a hyperplane. However, in our work, LinearSVC performs worse with an accuracy of 59%. The confusion matrix and Receiver Operating Characteristic (ROC) show a large amount of false-positives and false-negatives (Figures 19 and 20). There are 7843 benign packets, which are mistakenly classified as malware. In addition, 4387 malware packets were wrongly classified as benign. It can be inferred that the false-positives are higher than the false-negatives using the LinearSVC algorithm. To avoid overfitting and underfitting, we validated the performance of the model using the cross-validation technique with fold = 5. We applied the model on the training data and compared it to the performance of the model with testing data as shown in Table 5. The results show that the model does not suffer from overfitting or underfitting.

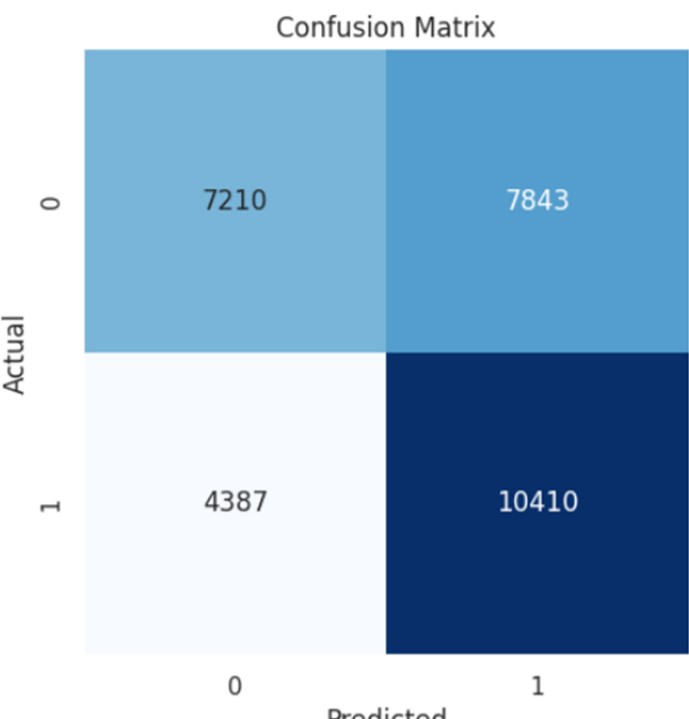

**Figure 19.** Confusion matrix for linearSVC.

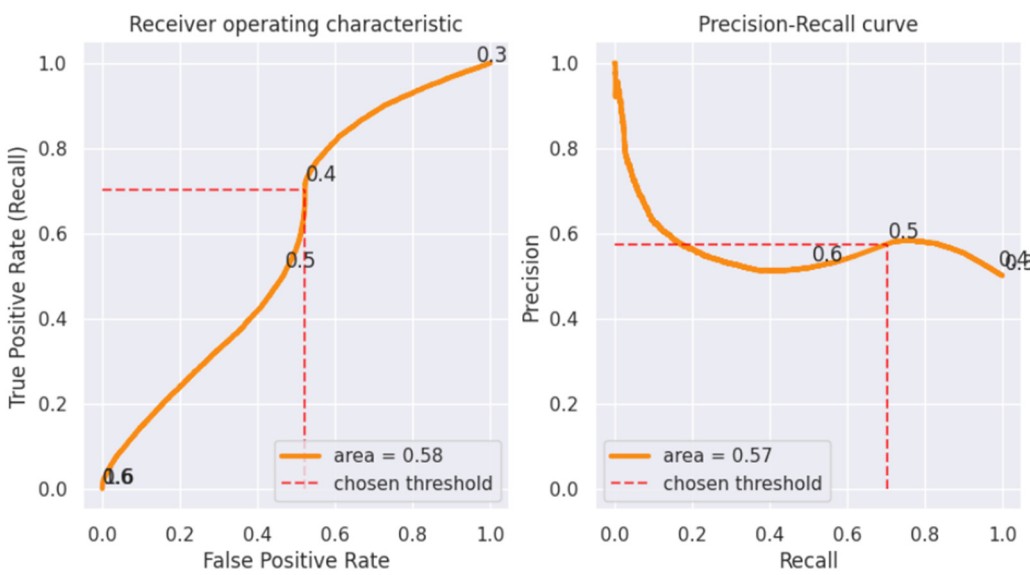

**Figure 20.** ROC curve for linearSVC.

**Table 5.** Cross-validation for linearSVC.

| Fold No. | Fold 1 | Fold 2 | Fold 3 | Fold 4 | Fold 5 | Model Accuracy |
|---|---|---|---|---|---|---|
| Performance | 0.654 | 0.651 | 0.653 | 0.653 | 0.645 | 0.655 |

### 4.2. Random Forest

Random Forest Classifier (FR) belongs to the decision tree algorithm family, which relies on ensemble methods to avoid overfitting and underfitting, which is common in traditional decision tree algorithms. The bagging methods are used to train RF by splitting the training data into sets, applying the decision tree for these sets, and for accumulating the results. Randomness and repetition of samples in RF is common, meaning a single instance may be used more than once due to recurrent sampling. The Random Forest Classifier shows better results than LinearSVC with an accuracy of 92%. However, the false-positive rate is fair with a total sample of 662 as shown in Figure 21. In contrast, the false-negative rate is higher with 1651 samples. Figure 22 shows the ROC curve for the RF algorithm. To validate our results, we used the cross-validation technique with fold = 5 as shown in Table 6.

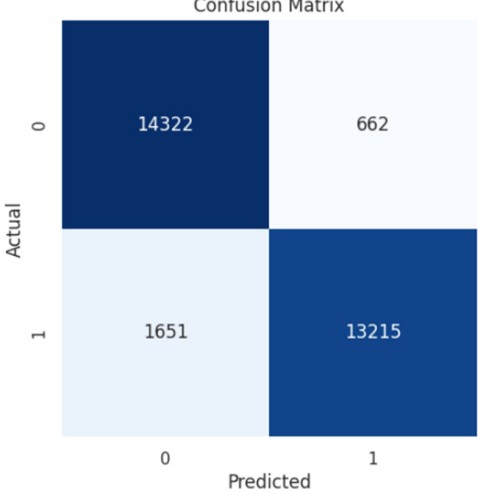

**Figure 21.** Confusion matrix for RF algorithm.

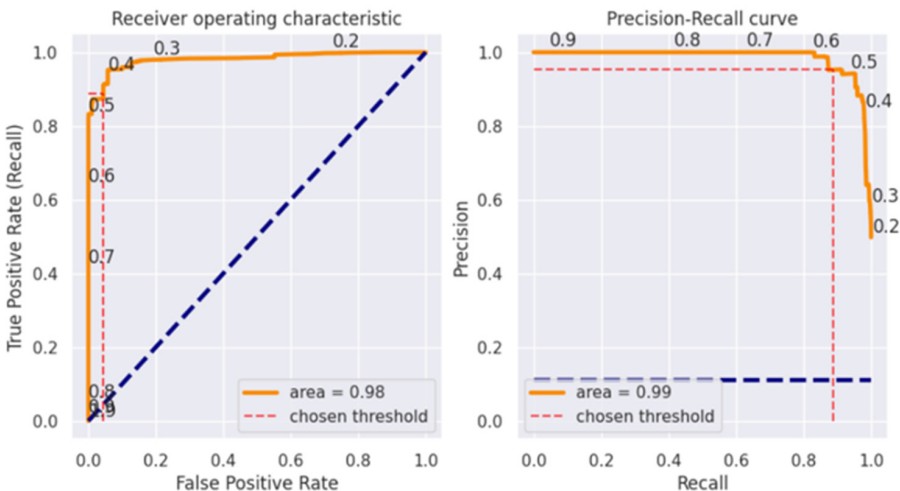

**Figure 22.** ROC curve for RF algorithm. Blue line represents area under the roc curve. To add it to the figure, the code must be updated. So, scientifically, the most significant part is to show the area in dark orange versus threshold.

**Table 6.** Cross-validation for RF.

| Fold No. | Fold 1 | Fold 2 | Fold 3 | Fold 4 | Fold 5 | Model Accuracy |
|----------|--------|--------|--------|--------|--------|----------------|
| Performance | 0.924 | 0.924 | 0.925 | 0.923 | 0.923 | 0.92 |

*4.3. Decision Tree*

Decision Tree (DT) is a famous algorithm in machine learning. It is suitable for classification and regression problems. The DT learning process is based on a sequence of comparisons between the data attributes (features), which result in more leaves that branch to the right or the left. When the learning reaches the end node, the decision is made based on the majority class in the leaf. A pre-learned threshold is set up to avoid the infinite process for learning. Compared to Neural Networks, DT performs well since it does not rely on gradient descent and the input normalization is not required. However, with image data, the neural networks outperform the DT. In our work, DT performs well with an accuracy of 98%. The number of false-positive rate is 347, while the false-negative rate is 302. The ROC curve infers the optimism of the decision tree algorithm. Figures 23 and 24 depict the confusion matrix and ROC curve for the DT, respectively. To validate the performance of DT, we apply the cross-validation technique with fold = 5 as shown in Table 7.

**Table 7.** Cross-validation for DT.

| Fold No. | Fold 1 | Fold 2 | Fold 3 | Fold 4 | Fold 5 | Model Accuracy |
|----------|--------|--------|--------|--------|--------|----------------|
| Performance | 0.985 | 0.985 | 0.985 | 0.984 | 0.987 | 0.987 |

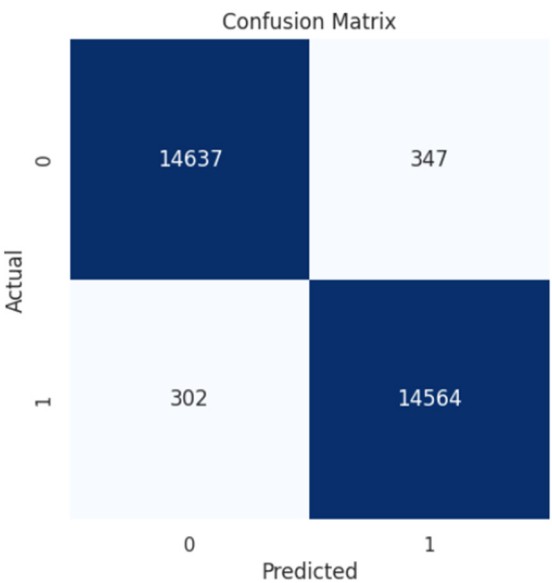

**Figure 23.** Confusion matrix for DT algorithm.

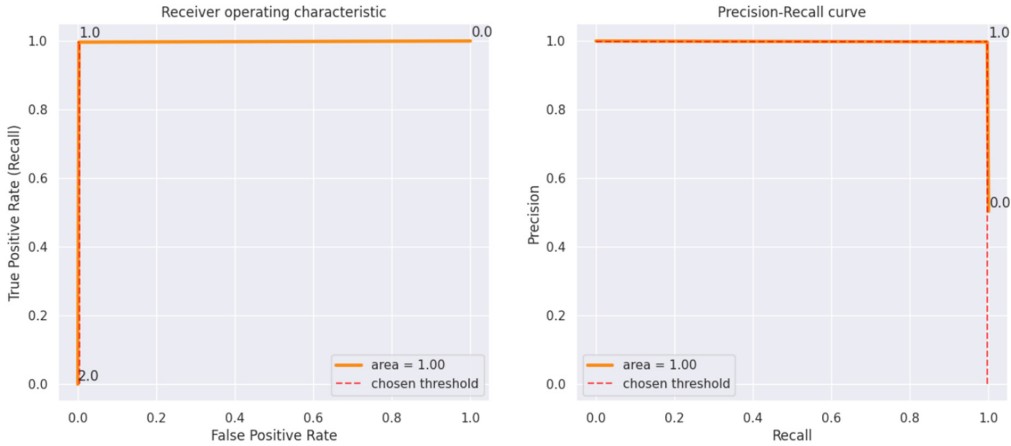

**Figure 24.** ROC curve for DT.

### 4.4. Naïve Byes

Naïve Byes (NB) originates from the statistical methods based on Byes Theorem. It is a machine learning classifier that is appropriate for classification problems. Moreover, it is fast, accurate, and performs well with large datasets. As the name implies, NB does not search for the relations between the features, assuming that each feature has an independent impact on the decision. The learning process is based on calculating the prior probability of a class label, then the likelihood for each feature for each class is calculated. The result is fed to the Byes formula to find the posterior probability. However, Naïve byes performs worse on our data and gains an accuracy of 70%. The false-negative rate is higher than the false-positive. The total number of samples of the false-negative rate is 6415 and 4612 samples for false-positive. We validated the results using the cross-validation technique as shown in Table 8. Figures 25 and 26 show the confusion matrix and ROC curve for Naïve Bayes performance.

**Table 8.** Cross-validation for NB.

| Fold No. | Fold 1 | Fold 2 | Fold 3 | Fold 4 | Fold 5 | Model Accuracy |
|---|---|---|---|---|---|---|
| Performance | 0.702 | 0.706 | 0.707 | 0.707 | 0.710 | 0.70 |

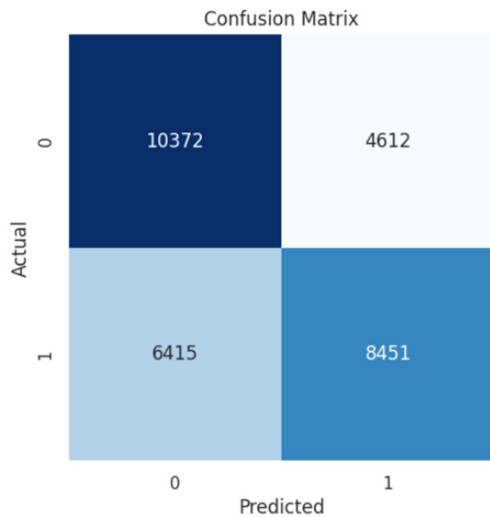

**Figure 25.** Confusion matrix for Naïve Byes.

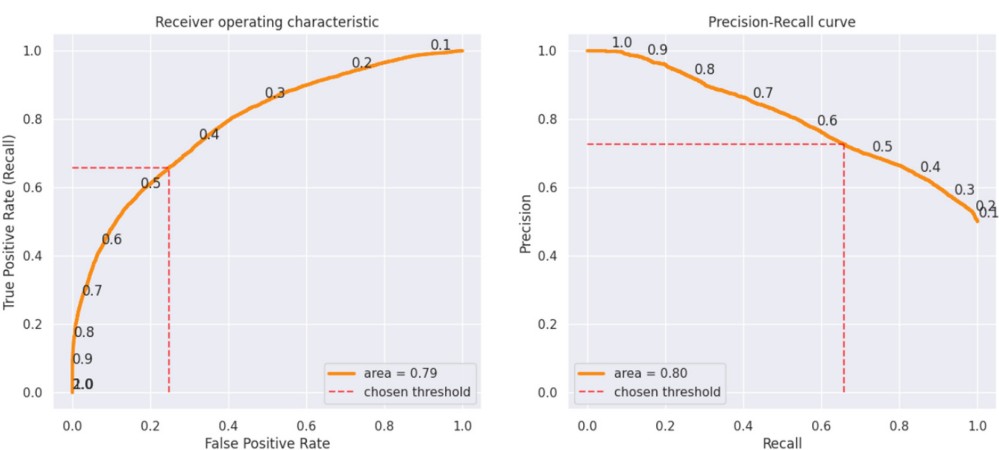

**Figure 26.** ROC curve for Naïve Byes.

From the results shown above, it can be inferred that the decision tree algorithm beats other algorithms in all the metrics used to calculate the performance of the selected algorithms. In the field of security, it is important to reduce the false-negative to the lowest value as the impact of single false-negative packet can cause large damage to the network. This is also shown by the performance of decision tree algorithm in our proposed model since only ~300 samples were misclassified as benign from a total of 30,000 samples that were used for testing. Figure 27 shows the calibration plot between the selected classifiers. It plots the DT performance with a significant similarity to the perfectly calibrated rather than the other algorithms. The mean predicted probability is plotted for each algorithm in Figure 28. The decision tree algorithm shows more impressive results than the other algorithms. Table 9 represents the performance of each algorithm in terms of accuracy, precision, recall, and F1-score.

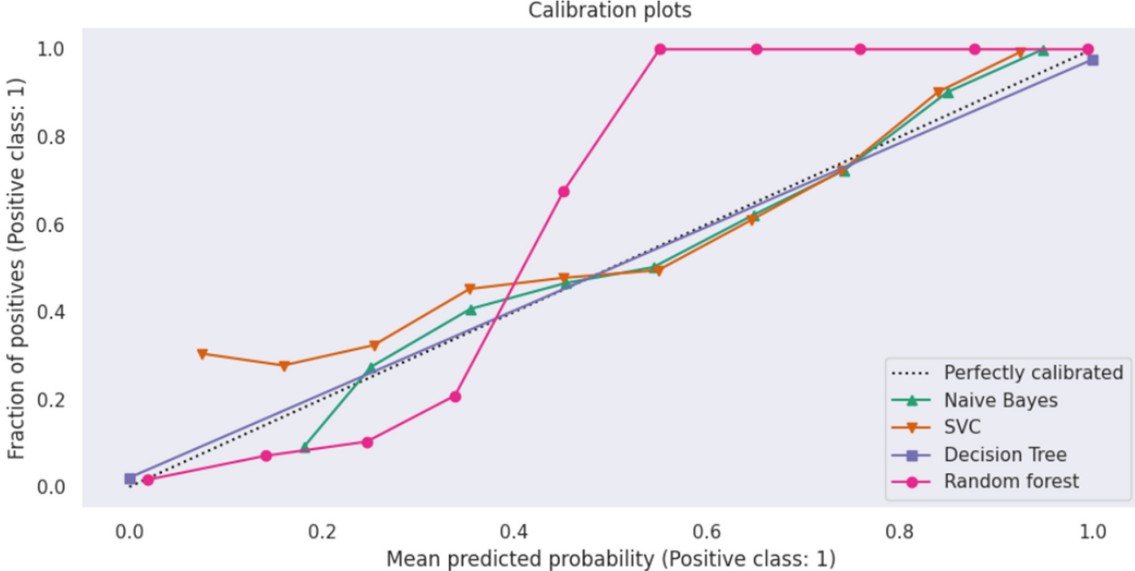

**Figure 27.** Calibration plot for the selected classifiers.

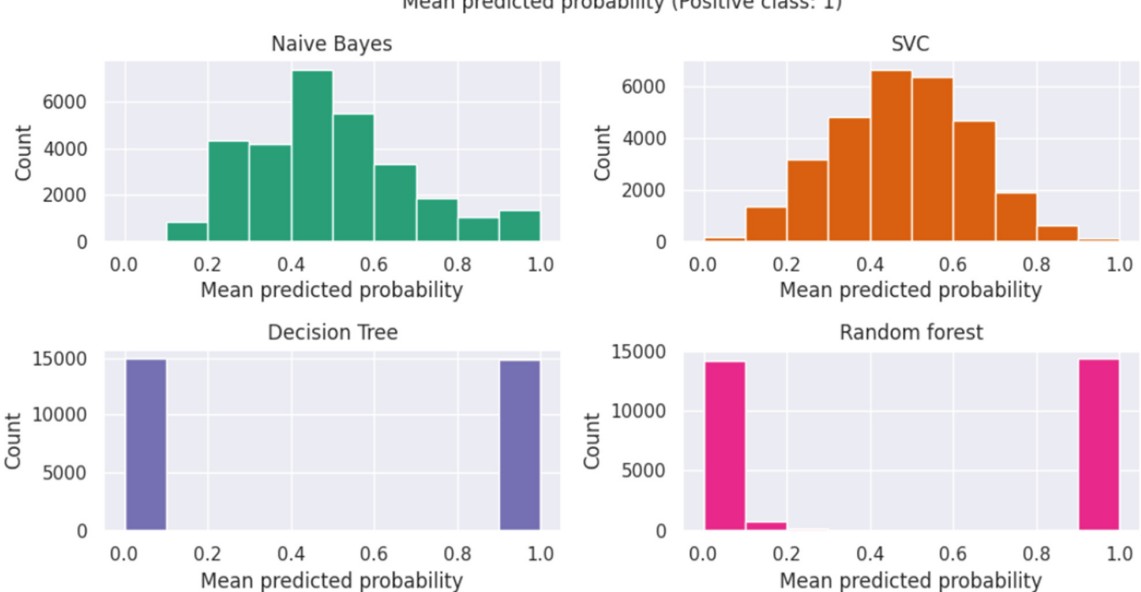

**Figure 28.** Mean predicted probability for the selected classifiers.

**Table 9.** Metrics for the selected algorithms.

| Model | Accuracy | Precision | Recall | F1-Score |
|---|---|---|---|---|
| LinearSVC | 59% | 62% | 48% | 54% |
| Decision Tree | 98% | 98% | 98% | 98% |
| Random Forest | 92% | 90% | 96% | 93% |
| Naïve Byes | 63% | 62% | 69% | 65% |

## 5. Discussion

The CoAP protocol suffers from the lack of research that can help in implementing and managing its security [6]. Several works employed DTLS to secure the CoAP protocol from different threats, including DDoS attacks. However, DTLS is not specially designed for constrained devices [6] and its heaviness renders it not suitable since they run on low power

and low energy. Moreover, DTLS is a cryptography-based protocol, and cryptography cannot detect attackers who use legal keys and act maliciously; therefore, DDoS attacks that originate from legitimate IPs may go undetected [21]. Consequently, the demand for new methods to secure CoAP against DDoS attacks is highly required since DTLS cannot protect CoAP from some of the internal and external attacks [22]. On the other hand, LSPWSN is also employed to secure CoAP messages. However, this protocol operates on the network layer, in contrast to CoAP which operates on the application layer. Moreover, while it is beneficial to mitigate the attack near the attack source, it is pertinent to have the detection system near the victim [7]. Therefore, this work outperforms DTLS and LSPWSN in terms of defending CoAP in its vicinity, while DTLS and LSPWSN can only protect CoAP by vetting the packets in the lower layers of IoT architecture, namely, the transport and network layers. Motivated by the assumption that it is beneficial to have the detection system near the victim [7], this work develops a model that can secure CoAP in its vicinity (the application layer). Another work [22] builds an anomaly-based detection method in the application layer to secure CoAP from DDoS attacks. However, this work collects those features from the protocols that work over the lower layers (IEEE 802.15.4 features, 6LoWPAN features, IPv6 features, and CoAP features) and gains an accuracy of 93%. The proposed method focuses only on the CoAP level features in the application layer and obtains an accuracy of 98% with the decision tree algorithm.

## 6. Conclusions

The proposed model defines a method to secure the CoAP in its vicinity from several DDoS attacks (duplication, modification, and intercepting of CoAP messages). The dataset used in this work has been extended from ~11,000 samples to 100,000 samples due to the lack of sufficient malware samples using GANs. The experimental performance was validated using a detection method in the same layer (application layer) and it is a significant work that fulfills the assumption of defending the resources in their vicinity. Therefore, this work discusses the different methods dedicated to detecting DDoS attacks that target CoAP. The proposed model can detect DDoS attacks with an accuracy of 98% with the decision tree algorithm. In the future, the performance of the proposed model can be improved by combining the dataset with other attacks, such as enumeration and amplification attacks, which will result in a coherent dataset for testing with new methods.

**Author Contributions:** Conceptualization, S.M.A. and A.A.-M.A.-G.; methodology, M.S.R. and M.R.; software, S.M.A.; validation, S.M.A., A.A.-M.A.-G. and M.S.R.; formal analysis, S.M.A.; investigation, A.A.-M.A.-G., M.S.R. and M.R.; resources, S.M.A.; data curation, S.M.A., A.A.-M.A.-G. and M.R.; writing—original draft preparation, S.M.A.; writing—review and editing, S.M.A. and M.R.; visualization; S.M.A.; funding acquisition, S.M.A.; supervision, A.A.-M.A.-G.; project administration, S.M.A. and M.R. All authors have read and agreed to the published version of the manuscript.

**Funding:** This research received no external funding.

**Data Availability Statement:** The dataset is publicly available at https://www.kaggle.com/datasets/salmeghlef/ddos-coap-dataset-cidad (accessed on 18 April 2023).

**Acknowledgments:** The authors gratefully acknowledge the Technical and Vocational Training Corporation (TVTC) for their support in performing this research. Also, many thanks for all the support and direction provided by the supervisors to produce this research. The authors gratefully acknowledge the support provided by the Faculty of Computing and Information Technology (FCIT), King Abdulaziz University (KAU), Jeddah, Saudi Arabia.

**Conflicts of Interest:** The authors declare no conflict of interest.

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
