# Peer review of "Application Layer-Based Denial-of-Service Attacks Detection against IoT-CoAP"

_electronics, doi:10.3390/electronics12122563_

Round 1

Reviewer 1 Report

This paper aims to  propose a method to detect DDoS attacks against CoAP in the application layer, whose contributions include:

1- Extending and balancing the CIDAD dataset using GANs. 

2- Focusing on the CoAP level features to ensure securing CoAP in its vicinity.

3- Build a machine learning model that can classify the benign against malware with an  accuracy of 98% using the decision tree algorithm. 

Overall, I think the work of this paper is important and meaningful. The paper is well written and structured. Besides, the origination structure of this paper is complete, which includes the presentations of theory itself, implementation, and experimental evaluation and result analysis. Therefore, I think this paper can be accepted after some minor revisions.

(1) Introduction.About 1.1 to 1.2 ,1)all content should be a separate chapter, where it is not appropriate.2)It is suggested that Chapter 2 be entitled Related Technologies, with subsequent chapters arranged in sequence.3)Reduce the irrelevant content in this section,the content is too redundant.

(2)The content of Section 2.3.1 should give the detailed design process of the proposed method rather than a literature review of existing methods.

(3)The different methods used in the experiment to analyze the detection results of denial of service attacks should be emphatically discussed.

other suggestion:

(1)In the abstract,the first appearance of CIDAD should be described.

(2)In Table 1,part of the Research Objective field has no date,so try to sort by date.

(3)The first line of the paragraph is indented by two characters. 

(4)Try to keep all images in the same format (including text size, centering, etc.), and some images have unclear text. Do not take screenshots.

(5)All table annotations should be at the top of the table.

(6)Figure 9 should be described as a table.

(7)In 2.1,"Next, we mention three of the discovered IoT botnets.  ",three should be four,please confirm.

(8)There are empty lines at the bottom of page 17.

Minor editing of English language required

Author Response

We would like to thank the reviewers and editors for providing an opportunity to revise the manuscript. We have studied these comments carefully and have made corresponding corrections that we hope will meet with your approval. In addition, as per the reviewer’s comment, we have improved the language quality of the manuscript and thoroughly proofread for grammatical as well as typographical errors. The revised text is mentioned in red font color.

 This paper aims to propose a method to detect DDoS attacks against CoAP in the application layer, whose contributions include:

1- Extending and balancing the CIDAD dataset using GANs. 

2- Focusing on the CoAP level features to ensure securing CoAP in its vicinity.

3- Build a machine learning model that can classify the benign against malware with an accuracy of 98% using the decision tree algorithm. 

Overall, I think the work of this paper is important and meaningful. The paper is well written and structured. Besides, the origination structure of this paper is complete, which includes the presentations of theory itself, implementation, and experimental evaluation and result analysis. Therefore, I think this paper can be accepted after some minor revisions.

  1. About 1.1 to 1.2 ,1) all content should be a separate chapter, where it is not appropriate.
  • Section 1.1, states the general structure for IoT network without discussing all the details of IoT network architecture. In Section 1.2, we discuss in detail one of the important parts of the IoT architecture which is IoT protocols. Since this work is focusing on securing one of the prevailing protocols that are used in IoT network, particularly CoAP, therefore, a detailed overview of the CoAP architecture is elaborated. Based on your suggestion, section 1.2 is embedded within section 1.1 in the updated version as follows:

1.1 IoT Overview

1.1.1 IoT Protocols

1.1.2 CoAP Architecture

1.1.2.1 Messaging Model

1.1.2.2 Request/Response Model

1.1.2.3 Message Format

1.1.2.4 Method Definitions

1.1.2.5 CoAP URIs

  1. It is suggested that Chapter 2 be entitled Related Technologies, with subsequent chapters arranged in sequence.3) Reduce the irrelevant content in this section, the content is too redundant.
  • Section 2 title is modified to “Related Technologies” and a hierarchical structure for the different methods is developed in the updated version. Indeed, in the research paper, we start reviewing the IoT defense mechanisms for mitigating DoS and DDoS attacks, then we proceed to discuss the same for CoAP specifically. Based on your suggestion, we proceed forward to the defense mechanisms for the CoAP security and grid of the defense mechanisms that are dedicated for IoT in general. This will result in removing the redundancy.
  1. Related Technologies

It is a complicated task to secure IoT from a variety of possible attacks [1]. Every layer in IoT architecture namely (Application layer, Network layer, and Infrastructure layer) is susceptible to different kinds of attacks. These attacks need to be identified and then develop security models to beat these attacks and ensure a secure IoT environment. Indeed, attacks show up due to some vulnerabilities present in the IoT environment. A vulnerability is incompetency that the attacker will exploit to penetrate the network. These vulnerabilities will result in threats to the IoT network if ignored, and consequently, it may lead to an attack. This work focuses on vulnerabilities related to DDoS attacks, so other vulnerabilities are out of the scope of this work.

2.1 DDoS attacks in IoT

Due to the huge amount of data flooding into the network, it may result in overwhelming the bandwidth, so the server will deny serving any request and will be inaccessible. This overflow of data is called a DDoS attack in which the legitimate user cannot access the server. Indeed, IoT which is considered a revolution in technology now can be a bane to it because IoT can contribute to DDoS attacks by adopting them negatively to create Botnets [9]. Botnets are defined as the process of infecting a massive number of IoT devices with malware to compromise these devices and make them under the control of an attacker who can trigger these devices to start an attack by dictating them to send massive traffics to the victim and consume the resources of that victim. Consequently, the infected IoT devices are called Bots. Normal IoT devices may be enforced to behave like a bot without the user's awareness. These bots are slaves and controlled by Master Bot Controller. This leads to making these botnets for sale and misuse nowadays. The Master Controller can attain some profits by selling their attack services. As a result, several variants of botnets have appeared in the market recently. Next, we mention four of the discovered IoT botnets.

A- Mirai

Mirai means the future in Japanese. It injected over 500.000 non-secured IoT devices, thus generating massive traffic to a target server with a flooding speed of 1 Tbps (Terabyte per second). The popularity of Mirai is gained from its irrefutable impact in 2016. It is designed to target Linux-running systems and some IoT devices, resulting in adding them as bots for the malware. Mirai targeted the famous Dyn DNS (French hosting provider) service in 2016, causing the internet inaccessible for major websites such as Twitter, Amazon, and Netflix. After that, IoT devices are exploited by different variants of Mirai and consequently, more DDoS attacks are launched.  It can be inferred that using IoT devices as botnets is one of the concerns in network security recently.

B- Wirex

Wirex is a botnet used to trigger a DDoS attack on multiple Content Delivery Networks (CDNs) and content providers. Wirex had infected an enormous number of Android devices having applications that behave as benign applications but were malware. Consequently, Google has banned some applications from the Play Store.

C- Reaper

This botnet is stronger than Mirai since the latter can penetrate devices with default credentials. This malware created a huge number of bots including Cisco routers and other brands by brutely adding them to its botnet. Reaper can exploit other vulnerabilities in IoT devices.

D- Torri

Torri is considered a new botnet nowadays [14]. It can target most of today’s recent computers, tablets, and smartphones due to its architectural design that has similar (64-bit), x86, etc.

2.2 CoAP Security Overview

This section introduces the DTLS binding for CoAP. Indeed, CoAP is equipped with the security demands such as keying materials and an access control list during the provisioning phase, or what so-called RawPublicKey mode. After RawPublicKey mode finishes, the IoT device should be in one of the following security modes:

A- NoSec Mode: No security protocol is engaged (DTLS disabled). Alternatively, assuming lower-layer protocol will implement the security mechanism; thus, messages are transferred with no security.

B- PresharedKey Mode: enabled DTLS. This mode has a list of pre-shared keys and there is a list of nodes that are assumed to engage in the communication for every key. For instance, in a significant scenario, every node has its key if it engages in CoAP communication. Contrarily, if a specific pre-shared key is shared with two entities or more, the entities are authenticated as a group by that key and would not be considered as specific peers anymore.

C- RawPublicKey Mode: enabled DTLS. This mode is adopted by the devices that required authentication, which uses the asymmetric key for each device to help to identify and communicate with these devices without a certificate.

D- Certificate Mode: DTLS is enabled.  In this mode, an asymmetric key pair with X.509 certificate is reserved for a given device. The certificate is validated by what is called Certification Authority (CA).

  1. The content of Section 2.3.1 should give the detailed design process of the proposed method rather than a literature review of existing methods.
  • 2.1 Proposed defense mechanisms for Securing CoAP against DDoS Attacks

2.2.1.1 DTLS for CoAP Security

Some researchers proposed that DTLS be implemented in CoAP for security purposes. Maleh et al., (2016) mentioned that Datagram Transport Layer (DTLS) handshake suffers from spoofing IP address DDoS attacks [15]. To face this issue, the DTLS handshake is expanded with a cookie exchange technique. The capability and threshold for receiving packets must be declared with the IP address to the server, then, the server reserves resources for new communication. Because of the costly energy of this technique, they moved it to Proxy AC Server with no energy constraints. Their method is depicted in Fig. 9, and they claim it reduces the resource occupancy of DTLS ROM by 23% compared to standard DTLS. Haroon A. et al. (2017) proposed an enhancement to DTLS to make it resistant to DDoS attacks [16]. They claim that their method can reduce the overhead of handshaking time, packet size, and energy consumption compared to other works. The authors’ system named E-lithe relies on Trusted Third Party (TTP) to reduce the overhead on the server-side. Compared to Lithe and other works, E-lithe outperforms others in terms of running time and reduced packet size. Later, this work was enhanced by Shruti et al., (2018) who claimed that their work outperforms E-Lithe [17] as depicted in Fig. 10. The authors’ work essentially focuses on the comparison between the payload of benign and malicious packets, defining a threshold, if the threshold is exceeded, then the source IP is blocked. They evaluate their work based on the malicious packet delivery ratio and legitimate packet drop ratio which outperforms the work done by Haroon et al.

                              Fig. 9. Cookie Exchange Technique by Maleh el al. [15]

Fig 10. Comparison between E-Lithe and Shruti et al. work [17]

2.2.1.2 SDN for CoAP Security

Alzahrani et al. (2020) implemented a software-defined networking scheme (depicted in Fig. 11) to authorize the messages over the CoAP protocol [18]. They argue that distributed approach in which IoT devices employ powerful gateways attached to them may be insufficient and making the access control decisions accomplished by the controller renders the security of CoAP messages more efficient and can help to avoid DDoS attacks.

Fig 11. SDN-based schema for CoAP message authorization [18]

2.2.1.3 Machine Learning for CoAP Security

Machine Learning is also proposed to detect and mitigate DDoS attacks against CoAP. Granjal et al., (2018) developed a framework that employs a threshold to mitigate DDoS attacks [19]. They define a limit for messages of CoAP, and after the limit is exceeded, extra messages will be dropped. The authors enhanced their work and proposed an anomaly-based detection system to protect the 6LoWPAN and CoAP protocol from DDoS attacks [20] depicted in Fig. 12. SVM is used as an ML-Classifier and gains an accuracy of 93%. However, the system generates a high false-positive rate of around 20%. Doshi et al., (2018) developed a machine learning pipeline (depicted in Fig. 13) that is employed on middleboxes such as routers or firewalls to detect IoT DDoS attacks [20]. They claim that IoT traffic is distinct from other traffic coming from other internet-connected devices because IoT traffic is repetitive and often communicates with a small finite of endpoints. After testing this method, it gains an accuracy of 99% using RF, KNN, and Neural Networks.

Fig. 12. Anomaly-based detection framework for CoAP DDoS attacks by Granjal et al. [18]

2.2.1.4 Other Methods for CoAP Security

Other works propose different methods to cope with DDoS attacks against CoAP. Anirudh et al. (2017) proposes a honeypot to lure the attacker and log his information for future verification or block purposes [21].

Fig 13. Machine Learning-based pipeline to mitigate CoAP DDoS attacks by Doshi et al. [20]

Fig 14. Honypot-based detection method for mitigating CoAP DDoS attacks by Anirudh et al. [21]

Detailed design for the proposed methods is included with supportive figures in the updated version.

  1. The different methods used in the experiment to analyze the detection results of denial-of-service attacks should be emphatically discussed.
  • We use Google Colab platform [24] to do the experiment and evaluate the performance for each model. Since our data is ~100,000 samples, the processing for these data and build a model require a high capability computing platform such as Google Colab platform. The steps to expertise the performance of the model are as follows:
  • Environment Setup: in this step, we import the data after cleaning and normalizing the data as previously discussed in sections 3.1 and 3.2.
  • Analyzing the data: we select only the features that are relevant to the label by calculating the Lasso regression and ANOVA analysis which results in choosing only 6 features from a total of 86 CoAP message features.
  • Data Preprocessing: in this phase, we handle the missing values by calculating the average of the column, and then we fill the missing values with the average. Some of the features contain large numbers, to face this issue and keep the model calculations simple, we normalize these values using max-min technique to bring the values between 0 and 1.
  • Model design: for each classifier, we run the experiment and validate the performance for each model using cross-validation technique to avoid overfitting and underfitting. We set up the fold to 5 and calculate the accuracy for each round and compare it to the model performance. No, overfitting nor underfitting is observed in our experiment.
  • Evaluate the performance: the performance for each model is calculated in terms of accuracy, precision, recall, and F1-score. In the next section, we elaborate the results of the model performance in detail.

Based on your comment, it is updated in the second version.

  1. In the abstract, the first appearance of CIDAD should be described.
  • The first appearance of CIDAD dataset was on Jul 11, 2022. It is included in the updated version.
  1. In Table 1, part of the Research Objective field has no date, so try to sort by date.
  • Table 1 is removed due to your recommendation for focusing on so close works for securing the CoAP instead of discussing the IoT security methods in general. Table 2. is sorted by date as follows:

Table 3: IoT-CoAP defense mechanisms

Limitation

Results

Methodology used

Research objective

The assumption of third party (Proxy AC server) is trustworthy all the time.

Low computation time by delegating all handshake to a third party

DTLS handshake is extended with a cookie exchange technique to check the authentication of a message.

Detect and mitigate DDoS attack against CoAP (2016)

DTLS is computationally heavy for IoT devices.

Energy consumption, reduced packet size and reduced running time outperforms similar works

Trusted Third Party (TTP) is used to avoid DoS attack and reduce overhead on the server side.

Secure DTLS for IoT

(2017)

Anomaly-based detection may result in high false positive

60% efficiency when a honeypot is implemented

Deploy a honeypot with two phases, first to log the anomalies and second to verify or block the client

Deploy a honeypot to detect DDoS attack

(2017)

Focuses on packet payload feature only

Evaluate malicious packet delivery ratio and legitimate packet drop ratio

Compare the payload of benign and malicious packet, define a threshold and if exceeded, the source IP is blocked

Detect DDoS against IoT (2018)

Susceptible to spoofing attack

Fair energy and memory consumption when running the proposed system.

Relies on threshold by limiting the incoming request to a fixed number and if exceeded, source request is blocked.

Preventation framework from intrusion and DoS attack (2018)

High false positive rate of 20%

SVM classifies the anomalies with accuracy of 93%

Employ anomaly-based detection system to protect CoAP from DoS attack.

Protect 6LoWPAN and CoAP from DoS attack

(2018)

Some IoT devices have different regular patterns but not distinct patterns

RF, KNN, Neural Networks gain about 99% accuracy

Machine learning DDoS detection framework

Detect IoT DDoS attack (2018)

Susceptible to other kinds of attacks

Compared to D-WARD, FR-WARD performs better in retransmition, duration and energy consumption.

Leverage the fast retransmit and flow control mechanism of TCP to retransmit benign packets at fastest rate and malicious packet at harmless rate.

Defend IoT against DDoS while maintaining benign traffic (2018)

Susceptible to sniffing attack

Burp suite tool is used to intercept the communication between the client and the server

Set up a client/server architecture to check if the communication between the two devices using CoAP is secure.

Test CoAP MITM attack which results in spoofing, sniffing and DoS attack (2019)

N/A

Decrease overhead to the controller and CoAP responses become faster

SDN-based approach is developed to authorize the messages of CoAP

Securing CoAP messages (2020)

It is fixed in the updated version.

  1. The first line of the paragraph is indented by two characters. 
  • Abstract: Internet of Things (IoT) is a massive network of tiny devices connected internally and to the internet. It is uniquely identified in the network (i.e., dedicated IP) and can share the information with other devices. However, the low power and low resources that distinguish IoT devices render them unsecure and targeted by different kinds of attacks since IoT devices can-not tolerate heavy security models. Also, due to the heavy nature of famous protocols such as HyperText Transport Protocol (HTTP), it is costly to be used with IoT devices, and alterna-tively, different lightweight protocols are implemented to fit IoT devices. One of the prevailing protocols used over IoT networks is the Constrained Application Protocol (CoAP). Therefore, CoAP is popular, and that makes it targeted by different types of attacks. One of the major attacks that target CoAP is distributed denial of service (DDoS) attacks. DDoS aims to over-whelm the resources of the target and make them unavailable to legitimate users. As a result, different kinds of methods were used to secure CoAP against DDoS attacks such as Datagram Transport Layer Security (DTLS) and Lightweight and Secure Protocol for Wireless Sensor Networks (LSPWSN). However, the existing models suffer from two issues: DTLS is not de-signed for constrained devices and is considered a heavy protocol. Besides, LSPWSN is working over the network layer, not in the application layer that CoAP works on. In this paper, we build a machine learning model that can detect the DDoS attacks against CoAP with an accuracy of 98%. The CIDAD dataset is extended from ~11000 to 100,000 samples using GANs be-because it has fewer samples of malware (less than 0.2% of the total dataset). Our model outperforms the existing models that target securing CoAP in the application layer and obtains 93% of accuracy.

Fixed in the updated version.

  1. Try to keep all images in the same format (including text size, centering, etc.), and some images have unclear text. Do not take screenshots.

The template of MDPI papers contain around 5cm on intendent space. So, we expand the images across the whole width of the document to keep the quality of the images and make it clear as follows:

Fixed in the updated version.

  1. All table annotations should be at the top of the table.
  • Table 1: Message header of CoAP

Version (V)

Type (T)

Token Length (TKL)

Code

MessageID

Token (if any)

Options (if any)

Payload (if any)

Table 2: IoT-CoAP defense mechanisms

Limitation

Results

Methodology used

Research objective

The assumption of third party (Proxy AC server) is trustworthy all the time.

Low computation time by delegating all handshake to a third party

DTLS handshake is extended with a cookie exchange technique to check the authentication of a message.

Detect and mitigate DDoS attack against CoAP (2016)

DTLS is computationally heavy for IoT devices.

Energy consumption, reduced packet size and reduced running time outperforms similar works

Trusted Third Party (TTP) is used to avoid DoS attack and reduce overhead on the server side.

Secure DTLS for IoT

(2017)

Anomaly-based detection may result in high false positive

60% efficiency when a honeypot is implemented

Deploy a honeypot with two phases, first to log the anomalies and second to verify or block the client

Deploy a honeypot to detect DDoS attack

(2017)

Focuses on packet payload feature only

Evaluate malicious packet delivery ratio and legitimate packet drop ratio

Compare the payload of benign and malicious packet, define a threshold and if exceeded, the source IP is blocked

Detect DDoS against IoT (2018)

Susceptible to spoofing attack

Fair energy and memory consumption when running the proposed system.

Relies on threshold by limiting the incoming request to a fixed number and if exceeded, source request is blocked.

Preventation framework from intrusion and DoS attack (2018)

High false positive rate of 20%

SVM classifies the anomalies with accuracy of 93%

Employ anomaly-based detection system to protect CoAP from DoS attack.

Protect 6LoWPAN and CoAP from DoS attack

(2018)

Some IoT devices have different regular patterns but not distinct patterns

RF, KNN, Neural Networks gain about 99% accuracy

Machine learning DDoS detection framework

Detect IoT DDoS attack (2018)

Susceptible to other kinds of attacks

Compared to D-WARD, FR-WARD performs better in retransmition, duration and energy consumption.

Leverage the fast retransmit and flow control mechanism of TCP to retransmit benign packets at fastest rate and malicious packet at harmless rate.

Defend IoT against DDoS while maintaining benign traffic (2018)

Susceptible to sniffing attack

Burp suite tool is used to intercept the communication between the client and the server

Set up a client/server architecture to check if the communication between the two devices using CoAP is secure.

Test CoAP MITM attack which results in spoofing, sniffing and DoS attack (2019)

N/A

Decrease overhead to the controller and CoAP responses become faster

SDN-based approach is developed to authorize the messages of CoAP

Securing CoAP messages (2020)

Table 3. CoAP Features Description

Feature

Description

Type

coap.mid

Message ID

Unsigned integer (2 bytes)

coap.opt.desc

Opt Desc

Character string

coap.opt.location_query

Location-Query

Character string

coap.opt.uri_host

Uri-Host

Character string

coap.retransmitted

Retransmitted

Label

coap.code

Code

Unsigned integer (1 byte)

Table3: Metrics for the selected algorithms

Model

Accuracy

Precision

Recall

F1-Score

LinearSVC

59%

62%

48%

54%

Decision Tree

98%

98%

98%

98%

Random Forest

92%

90%

96%

93%

Naïve Byes

63%

62%

69%

65%

Fixed in the updated version.

  1. Figure 9 should be described as a table.

Table 1: Message header of CoAP

Version (V)

Type (T)

Token Length (TKL)

Code

MessageID

Token (if any)

Options (if any)

Payload (if any)

Added in the updated version.

  1. In 2.1,"Next, we mention three of the discovered IoT botnets.  ", three should be four, please confirm.
  • It is four and corrected in the updated version.

  1. There are empty lines at the bottom of page 17.
  • Fixed in the updated version.

  1. English language required.
  • As per the reviewer’s comment, we have improved the language quality of the manuscript and thoroughly proofread for grammatical as well as typographical errors.

We thank the reviewer for the positive comments.

Reviewer 2 Report

The work entitled “Application Layer-Based Denial-of-Service Attacks Detection Against IoT-CoAP” by Sultan Almeghlef and collaborators describes the generation of a dataset using Generative Adversarial Network (GAN) to train a model to identify benign IoT communications from malware.

The work is well written, but some limitations are found in the description of their work. In particular:

Major concerns

1.     Classical machine learning (ML) techniques are used, as opposed to deep learning ones, when there is a set of features that are related to the label to be classified. The basis for that is because classical ML techniques may not transform the features representation, instead they try to find the border splitting the different classes. In that sense, the authors note that there are 86 features associated with CoAP. Which are those 86 features? Are there any of those features that are more likely to be associated with malwares? If so, a protocol should enforce those features to be received and train an algorithm to detect malwares from those features. In that sense, the authors use too much space to describe the basis of the IoT, but failed to describe this aspect that is fundamental to their work. Is it possible to anticipate that some features are more likely associated with a malware? If the answer is no, then, deep learning should be preferred for classifying these data. Based on the 10 features used by the authors and the low Pearson correlation reported with malware/benign labels, it is not anticipated that such features are related to malwares. It may seem that the classification observed is the consequence of the GAN procedure used. To address this problem, it is important to use alternatives procedures to generate train/test sets, namely undersampling and oversampling. Undersampling uses the larger set (in this case, benign) to randomly select a sample whose size would be equal to the small one (malware); in this case, the authors may test whether the larger set has outliers that may prove difficult for the classification task, hence showing potential failures in their classification. GAN is an example of oversampling, yet there are other approaches (e.g., SMOTE); using alternative oversampling techniques may assist the authors to validate their results. While this is not common practice, given the nature of this problem, it is advisable for authors to do this.

2.     The best methods to classify the GAN generated data was RF and DT, but LinearSVM failed to classify these data. This last result indicates that benign and malware generated data cannot be separated by a hyperplane. Considering that, it is possible that RF and DT succeeded because there are many small sets of malwares, hence preventing a simple hyperplane to separate benign from malwares and suggesting possibly overfitting. While authors describe in their methods to have used 70% of the data in training and 30% of the data in testing, only one performance is reported (e.g., tables 16, 18 and 20). It is not clear whether these results correspond with the training or testing, from line 670 it seems that the authors report only the test results. Showing both efficiencies for training and testing will allow authors to rule out the possible overfitting of the reported algorithms. Additionally, the authors may use under sampling and alternative oversampling techniques to generate new data to test for overfitting.

3.     The authors make a small discussion of their results with previous reports on this same subject. In the discussion, they only compare their efficiencies with reference 33 and conclude their method is better because they achieved 98% instead of 93% of efficiency. Based only on that percentage, one can conclude their method is not better than the work reported in reference 34, where they report to have achieved 99% using RF as well. A fair comparison should include datasets and efficiencies. It is recommended for authors to make a better comparison and discussion of their results.

Minor concerns

1.     Lines 80-91 are repetitive, may be eliminated.

2.     Line 132, M2M, please define it.

3.     Line 546, “We used to use Generative Adversarial Network (GAN)…” should be “We used the Generative Adversarial Network (GAN)…”

4.     Line 549, RMSE, please define.

5.     Line 566, "decent", should use a more precise way to describe these 10 features.

6.     Lines 567-568, which label? benign or malware? Or both? Please specify.

Some minor errors were identified in the text. Some of them are commented in the notes for authors. It is highly advisable for authors to ask a native-speaking person to review their work.

Author Response

We would like to thank the reviewers and editors for providing an opportunity to revise the manuscript. We have studied these comments carefully and have made corresponding corrections that we hope will meet with your approval. In addition, as per the reviewer’s comment, we have improved the language quality of the manuscript and thoroughly proofread for grammatical as well as typographical errors. The revised text is mentioned in red font color.

The work entitled “Application Layer-Based Denial-of-Service Attacks Detection Against IoT-CoAP” by Sultan Almeghlef and collaborators describes the generation of a dataset using Generative Adversarial Network (GAN) to train a model to identify benign IoT communications from malware.

The work is well written, but some limitations are found in the description of their work. In particular:

Major concerns

  1. Classical machine learning (ML) techniques are used, as opposed to deep learning ones, when there is a set of features that are related to the label to be classified. The basis for that is because classical ML techniques may not transform the features representation, instead they try to find the border splitting the different classes. In that sense, the authors note that there are 86 features associated with CoAP. Which are those 86 features? Are there any of those features that are more likely to be associated with malwares? If so, a protocol should enforce those features to be received and train an algorithm to detect malwares from those features. In that sense, the authors use too much space to describe the basis of the IoT, but failed to describe this aspect that is fundamental to their work. Is it possible to anticipate that some features are more likely associated with a malware? If the answer is no, then, deep learning should be preferred for classifying these data. Based on the 10 features used by the authors and the low Pearson correlation reported with malware/benign labels, it is not anticipated that such features are related to malwares. It may seem that the classification observed is the consequence of the GAN procedure used. To address this problem, it is important to use alternatives procedures to generate train/test sets, namely undersampling and oversampling. Undersampling uses the larger set (in this case, benign) to randomly select a sample whose size would be equal to the small one (malware); in this case, the authors may test whether the larger set has outliers that may prove difficult for the classification task, hence showing potential failures in their classification. GAN is an example of oversampling, yet there are other approaches (e.g., SMOTE); using alternative oversampling techniques may assist the authors to validate their results. While this is not common practice, given the nature of this problem, it is advisable for authors to do this.

  • Based on Wireshark documentation, CoAP communication has a total of 86 features, most of which are optional features that may or may not appear in the communication process. The table below describes the 86 associated with CoAP.

Field name

Description

Type

Versions

coap.block

Block

Frame number

3.4.0 to 4.0.5

coap.block.count

Block count

Unsigned integer (32 bits)

3.4.0 to 4.0.5

coap.block.error

Block defragmentation error

Frame number

3.4.0 to 4.0.5

coap.block.multiple_tails

Block has multiple tails

Boolean

3.4.0 to 4.0.5

coap.block.overlap

Block overlap

Boolean

3.4.0 to 4.0.5

coap.block.overlap.conflicts

Block overlapping with conflicting data

Boolean

3.4.0 to 4.0.5

coap.block.reassembled.in

Reassembled in

Frame number

3.4.0 to 4.0.5

coap.block.reassembled.length

Reassembled block length

Unsigned integer (32 bits)

3.4.0 to 4.0.5

coap.block.too_long

Block too long

Boolean

3.4.0 to 4.0.5

coap.block_length

Block Length

Unsigned integer (32 bits)

3.4.0 to 4.0.5

coap.block_payload

Block Payload

Byte sequence

3.4.0 to 4.0.5

coap.blocks

Blocks

Label

3.4.0 to 4.0.5

coap.code

Code

Unsigned integer (8 bits)

1.6.0 to 4.0.5

coap.invalid_option_number

Invalid Option Number

Label

1.12.0 to 4.0.5

coap.invalid_option_range

Invalid Option Range

Label

1.12.0 to 4.0.5

coap.length

Length

Unsigned integer (32 bits)

3.2.0 to 4.0.5

coap.mid

Message ID

Unsigned integer (16 bits)

1.12.0 to 4.0.5

coap.ocount

Opt Count

Unsigned integer (8 bits)

1.10.0 to 1.10.14

coap.opt.accept

Accept

Character string

1.8.0 to 4.0.5

coap.opt.block_mflag

More Flag

Unsigned integer (8 bits)

1.6.0 to 4.0.5

coap.opt.block_number

Block Number

Unsigned integer (32 bits)

1.6.0 to 4.0.5

coap.opt.block_size

Encoded Block Size

Unsigned integer (8 bits)

1.6.0 to 4.0.5

coap.opt.ctype

Content-type

Character string

1.6.0 to 4.0.5

coap.opt.delta

Opt Delta

Unsigned integer (8 bits)

1.6.0 to 4.0.5

coap.opt.delta_ext

Opt Delta extended

Unsigned integer (16 bits)

1.12.0 to 4.0.5

coap.opt.desc

Opt Desc

Character string

1.10.0 to 4.0.5

coap.opt.end_marker

End of options marker

Unsigned integer (8 bits)

1.12.0 to 4.0.5

coap.opt.etag

Etag

Byte sequence

1.6.0 to 4.0.5

coap.opt.hop_limit

Hop Limit

Unsigned integer (8 bits)

3.4.0 to 4.0.5

coap.opt.if_match

If-Match

Byte sequence

1.8.0 to 4.0.5

coap.opt.if_none_match

If-None-Match

Byte sequence

1.8.0 to 1.8.15

coap.opt.jump

Opt Jump

Unsigned integer (8 bits)

1.10.0 to 1.10.14

coap.opt.length

Opt Length

Unsigned integer (8 bits)

1.6.0 to 4.0.5

coap.opt.length_ext

Opt Length extended

Unsigned integer (16 bits)

1.12.0 to 4.0.5

coap.opt.location

Location

Character string

1.6.0 to 1.6.16

coap.opt.location_path

Location-Path

Character string

1.8.0 to 4.0.5

coap.opt.location_query

Location-Query

Character string

1.8.0 to 4.0.5

coap.opt.max_age

Max-age

Unsigned integer (32 bits)

1.6.0 to 4.0.5

coap.opt.name

Opt Name

Character string

1.10.0 to 4.0.5

coap.opt.object_security_expand

Expanded Flag Byte

Boolean

2.6.0 to 3.2.18

coap.opt.object_security_kid

Key ID

Byte sequence

2.6.0 to 4.0.5

coap.opt.object_security_kid_context

Key ID Context

Byte sequence

2.6.0 to 4.0.5

coap.opt.object_security_kid_context_len

Key ID Context Length

Unsigned integer (8 bits)

2.6.0 to 4.0.5

coap.opt.object_security_kid_context_present

Key ID Context Present

Boolean

2.6.0 to 4.0.5

coap.opt.object_security_kid_present

Key ID Present

Boolean

3.0.0 to 4.0.5

coap.opt.object_security_non_compressed

Non-compressed COSE message

Boolean

2.6.0 to 3.2.18

coap.opt.object_security_piv

Partial IV

Byte sequence

2.6.0 to 4.0.5

coap.opt.object_security_piv_len

Partial IV Length

Unsigned integer (8 bits)

2.6.0 to 4.0.5

coap.opt.object_security_reserved

Reserved

Boolean

3.4.0 to 4.0.5

coap.opt.object_security_signature

Signature Present

Boolean

2.6.0 to 3.2.18

coap.opt.observe

Observe

Unsigned integer (32 bits)

2.0.0 to 4.0.5

coap.opt.payload_desc

Payload Desc

Character string

1.10.0 to 2.2.17

coap.opt.proxy_scheme

Proxy-Scheme

Character string

2.0.0 to 4.0.5

coap.opt.proxy_uri

Proxy-Uri

Character string

1.8.0 to 4.0.5

coap.opt.size1

Size1

Unsigned integer (32 bits)

2.0.0 to 4.0.5

coap.opt.subscr_lifetime

Lifetime

Unsigned integer (32 bits)

1.6.0 to 1.12.13

coap.opt.token

Token

Character string

1.6.0 to 1.10.14

coap.opt.unknown

Unknown

Byte sequence

1.10.0 to 4.0.5

coap.opt.uri_auth

Uri-Authority

Character string

1.6.0 to 1.6.16

coap.opt.uri_host

Uri-Host

Character string

1.8.0 to 4.0.5

coap.opt.uri_path

Uri-Path

Character string

1.6.0 to 4.0.5

coap.opt.uri_path_recon

Uri-Path

Character string

2.4.0 to 4.0.5

coap.opt.uri_port

Uri-Port

Unsigned integer (16 bits)

1.8.0 to 4.0.5

coap.opt.uri_query

Uri-Query

Character string

1.6.0 to 4.0.5

coap.optcount

Option Count

Unsigned integer (8 bits)

1.6.0 to 1.8.15

coap.option_length_bad

Option length bad

Label

1.12.0 to 4.0.5

coap.option_object_security_bad

Invalid Object-Security Option Format

Label

2.6.0 to 3.2.2

coap.option_oscore_bad

Invalid OSCORE Option Format

Label

3.2.3 to 4.0.5

coap.oscore_kid

OSCORE Key ID

Byte sequence

2.6.0 to 4.0.5

coap.oscore_kid_context

OSCORE Key ID Context

Byte sequence

2.6.0 to 4.0.5

coap.oscore_piv

OSCORE Partial IV

Byte sequence

2.6.0 to 4.0.5

coap.payload

Payload

Character string

1.12.0 to 4.0.5

coap.payload_desc

Payload Desc

Character string

2.4.0 to 4.0.5

coap.payload_length

Payload Length

Unsigned integer (32 bits)

2.4.0 to 4.0.5

coap.request_first_in

Retransmission of request in

Frame number

3.2.0 to 4.0.5

coap.response_first_in

Retransmission of response in

Frame number

3.2.0 to 4.0.5

coap.response_in

Response In

Frame number

2.2.0 to 4.0.5

coap.response_time

Response Time

Time offset

2.2.0 to 4.0.5

coap.response_to

Request In

Frame number

2.2.0 to 4.0.5

coap.retransmitted

Retransmitted

Label

3.2.0 to 4.0.5

coap.tid

Transaction ID

Unsigned integer (16 bits)

1.6.0 to 1.10.14

coap.token

Token

Byte sequence

1.12.0 to 4.0.5

coap.token_len

Token Length

Unsigned integer (8 bits)

1.12.0 to 4.0.5

coap.type

Type

Unsigned integer (8 bits)

1.6.0 to 4.0.5

coap.unknown_option_number

Unknown Option Number

Label

3.2.5 to 4.0.5

coap.version

Version

Unsigned integer (8 bits)

1.6.0 to 4.0.5

As the name of the feature indicates (option, opt), it means the feature is optional to show up in the communication process. When we ran the experiment, and during the feature extraction phase, our observation is that from the 86 features, only 24 features are coherent and contain values for each communication process, and three optional features (coap.opt.desc, coap.opt.location_query, coap.opt.max_age) contain more than 90% of values from the communication process,  while others have massive missing values. It is not appropriate to fill out the missing values with the average because the number of missing values against the existing ones is very large. Consequently, we grid of all the optional features except the three mentioned above and focus only on the CoAP Non-Optional features which include 24 features as shown in Table 2 and the other three optional features.  

Table 2. CoAP Non-Optional Features

Field name

Description

Type

Coap.block_length

Block length

Unsigned Integer (4 bytes)

Coap.blocks

Block

Label

Coap.code

Code

Unsigned Integer

Coap.token_len

Token length

Unsigned Integer (1 byte)

Coap.type

Type

Unsigned Integer (1 byte)

Coap.block

Block

Frame number

Coap.block. count

Block count

Unsigned Integer (4 bytes)

Coap.block. error

Block defragmentation error

Frame number

Coap.block. multiple_tails

Block has multiple tails

Boolean

Coap.block. overlap

Block overlap

Boolean

Coap.block.overlap.conflicts

Block overlapping with conflicting data

Boolean

Coap.block.reassembled.in

Reassembled in

Frame number

coap.block.reassembled.length

Reassembled block length

Unsigned integer (4 byte)

coap.block.too_long

Block too long

Boolean

coap.length

Length

Unsigned integer (4 byte)

coap.oscore_kid

OSCORE Key ID

Byte sequence

coap.oscore_kid_context

OSCORE Key ID context

Byte sequence

coap.oscore_piv

OSCORE Partial IV

Byte sequence

coap.payload

Payload

Character string

coap.payload_desc

Payload Desc

Character string

coap.payload_length

Payload Length

Unsigned integer (4 byte)

coap.request_first_in

Retransmission of request in

Frame number

coap.response_in

Response in

Frame number

coap.retransmitted

Retransmitted

Label

After selecting the 24 non-optional features, we proceed to find the most relevant features to the label. The complete dataset is used (benign and malware together) to find the most relevant features to the label using Lasso Regression and ANOVA analysis. The threshold is set to ±0.30 and our observation was only 6 features have scored more than the threshold. Regarding if the selected features are scoring better due to association with malware, our observation does not show this result and most of the selected features have convergent values for both classes (benign and malware) as shown in the sample of the dataset in Table 3.

Table 3. Sample of CIDAD dataset

coap.mid

coap.code

coap.opt.desc

coap.opt.location_query

coap.retransmitted

coap.token

class

0.72071268

0.94586302

0.09609074

0.11852746

0.6457001

0.44520495

1

0.06378486

0.05733755

0.19273773

0.06178549

0.99847181

0.39920222

1

0.22397982

0.03890724

0.01806011

0.01064374

0.74956856

0.62831367

0

0.35310734

0.95677138

0.03130444

0.01226962

0.54031596

0.32082396

0

0.11846928

0.07303984

0.03400905

0.99913829

0.97845709

0.66386946

1

0.82660099

0.04260864

0.0347903

0.03692305

0.94372874

0.34955944

0

0.78141243

0.16411787

0.05012246

0.13920499

0.86151786

0.62702198

1

0.99822554

0.00013203

0.98110964

0.94997002

0.40334718

0.73456738

0

0.75422632

0.02346656

0.89522515

0.13964064

0.15678872

0.54074526

1

0.41376686

0.02879173

0.91810143

0.19033603

0.75274199

0.84263953

1

Regarding connecting our work to the basis of IoT, we remove the part that focuses on reviewing the defense mechanisms against DDoS for IoT in general and we focus only on the defense mechanisms that are dedicated for securing CoAP from DDoS attacks. To validate our work, we implement cross validation technique with fold=5, our observation shows that the model performance on the training data is so close the test data. As a result, our model is not suffering from underfitting or overfitting. To check whether the classifier rely on the GAN samples, in our work, both benign and malware are extended using GANs. The original number of benign was ~10,000 against 289 only of malware. So, we extend both to ~50,000 for each class and vet both sets for outliers and missing values. Our observation is that the traditional statical methods namely Lasso regression and ANOVA analysis suggest the most relevant features to the label.

  1. The best methods to classify the GAN generated data was RF and DT, but LinearSVM failed to classify these data. This last result indicates that benign and malware generated data cannot be separated by a hyperplane. Considering that, it is possible that RF and DT succeeded because there are many small sets of malwares, hence preventing a simple hyperplane to separate benign from malwares and suggesting possibly overfitting. While authors describe in their methods to have used 70% of the data in training and 30% of the data in testing, only one performance is reported (e.g., tables 16, 18 and 20). It is not clear whether these results correspond with the training or testing, from line 670 it seems that the authors report only the test results. Showing both efficiencies for training and testing will allow authors to rule out the possible overfitting of the reported algorithms. Additionally, the authors may use under sampling and alternative oversampling techniques to generate new data to test for overfitting.

  • RF and DT performs well in classifying the benign against the malware while the dataset is balanced to 50% for each class. We validate our results on both training and testing data to check for overfitting. This part is discussed in detail in the updated version.

Major concerns

  1. The authors make a small discussion of their results with previous reports on this same subject. In the discussion, they only compare their efficiencies with reference 33 and conclude their method is better because they achieved 98% instead of 93% of efficiency. Based only on that percentage, one can conclude their method is not better than the work reported in reference 34, where they report to have achieved 99% using RF as well. A fair comparison should include datasets and efficiencies. It is recommended for authors to make a better comparison and discussion of their results.

            Our motivation in this research is that we are focusing in securing CoAP in its perimeter due to the assumption that it is beneficial to secure the resource in its vicinity while the mitigation should be closer to the attacker. For this reason, we claim that our work outperforms other works in terms of the dataset used in this research which contains only the CoAP level packet in the application layer while not relying on the down layers to vet the CoAP messages such as transport layer in IoT network architecture. The comparison is updated in the second version.

  1. Lines 80-91 are repetitive, may be eliminated.
  • IoT Overview

The Internet of Things inspires the existence of tiny object devices so they can send and receive data with little users’ intervention [9]. As per the anticipation of some research, 75 billion IoT devices will take place in the communication technology environment by 2025 [2].  IoT consists of three layers (as depicted in Fig.1): the perception layer that represents the physical sensors and actuators, the network layer that enables device-to-device and device-to-cloud communication, and the application layer that delivers the services to other devices or humans [2]. IoT architecture can be extended to have two more layers, namely, MAC and adaptation layer resulting in what the so-called five-layer architecture [10]. Due to the heterogeneity aspect of IoT devices that will interact with other devices, and different IoT protocols are developed to enable the interactions between IoT devices. Consequently, the Institute of Electrical and Electronics Engineers (IEEE) and the Internet Engineering Task Force (IETF) are developing standardized protocols. Fig.2 depicts the layers and protocols deployed for each layer based on the five-layer architecture of the IoT network. After reviewing the IoT architecture, the protocols related to the IoT will be discussed with more focus on the CoAP protocol. CoAP prevails among other protocols in the application layer because it is a lightweight protocol and fits for constrained devices [5]. However, CoAP and many other protocols are susceptible to DDoS attacks. DDoS attacks target different layers and many protocols rendering the service unavailable for legitimate users.

  • Fixed in the updated version.

  1. Line 132, M2M, please define it.

  • Machine-to-Machine communication.

Corrected in the updated version.

  1. Line 546, “We used to use Generative Adversarial Network (GAN)…” should be “We used the Generative Adversarial Network (GAN)…”

  • We used Generative Adversarial Network (GAN)…

Corrected in the updated version.

  1. Line 549, RMSE, please define.
  • Root Mean Square Error (RMSE).

Corrected in the updated version.

  1. Line 566, "decent", should use a more precise way to describe these 10 features.
  • Only 10 features have fewer missing values (less than 10% of the column data) in the dataset.

Fixed in the updated version.

  1. Lines 567-568, which label? benign or malware? Or both? Please specify.
  • Then, we use the Pearson Correlation method formula (1) to get the relevant features to the labels (benign and malware).

Corrected in the updated version.

  • As per the reviewer’s comment, we have improved the language quality of the manuscript and thoroughly proofread for grammatical as well as typographical errors.

We thank the reviewer for the positive comments.

Reviewer 3 Report

Generally speaking, the paper, "Application Layer-Based Denial-of-Service Attacks Detection Against IoT-CoAP," addresses a critical issue in IoT security, the vulnerability of Constrained Application Protocol (CoAP) to Distributed Denial of Service (DDoS) attacks. This work is significant given the rising prominence of IoT devices and the inherent limitations of these devices in terms of power and resources that make them susceptible to security threats.

The authors effectively critique current security measures such as Datagram Transport Layer Security (DTLS) and Lightweight and Secure Protocol for Wireless Sensor Networks (LSPWSN), which fail to adequately address these issues due to their resource-intensive nature or inappropriate layer of operation. Their proposed solution, a machine learning model for detecting DDoS attacks against CoAP, is promising, boasting a reported accuracy of 98%. This performance is impressive, particularly given that it only considers CoAP-level features, thus improving on existing models by focusing on the application layer where CoAP operates. The use of Generative Adversarial Networks (GANs) to extend the CIDAD dataset from ~11,000 to 100,000 samples indicates a sophisticated approach to the data scarcity problem.

However, the paper could benefit from a more in-depth discussion of the machine learning model itself. Details about the specific type of model used, its training process, and feature selection would be beneficial for understanding the solution more thoroughly and potentially replicating the study.

In the conclusion, the authors suggest future work involving the deployment of the model and its combination with other types of attacks for comprehensive testing. This provides a clear direction for subsequent research. Overall, the paper presents a promising solution to a pertinent issue in IoT security, though additional details about the machine learning model would enhance its impact and usability.

Last but not least, the authors should elaborate more on the resource (hardware and time) consumption spent on the GANs models, and extensively comment on the complexity.

Also, the authors are suggested to recheck the plots well-scaled with enough resolution to illustrate.

Well handled

Author Response

We would like to thank the reviewers and editors for providing an opportunity to revise the manuscript. We have studied these comments carefully and have made corresponding corrections that we hope will meet with your approval. In addition, as per the reviewer’s comment, we have improved the language quality of the manuscript and thoroughly proofread for grammatical as well as typographical errors. The revised text is mentioned in red font color.

  1. Generally speaking, the paper, "Application Layer-Based Denial-of-Service Attacks Detection Against IoT-CoAP," addresses a critical issue in IoT security, the vulnerability of Constrained Application Protocol (CoAP) to Distributed Denial of Service (DDoS) attacks. This work is significant given the rising prominence of IoT devices and the inherent limitations of these devices in terms of power and resources that make them susceptible to security threats.
  • We thank the reviewer for the positive comments.

  1. The authors effectively critique current security measures such as Datagram Transport Layer Security (DTLS) and Lightweight and Secure Protocol for Wireless Sensor Networks (LSPWSN), which fail to adequately address these issues due to their resource-intensive nature or inappropriate layer of operation. Their proposed solution, a machine learning model for detecting DDoS attacks against CoAP, is promising, boasting a reported accuracy of 98%. This performance is impressive, particularly given that it only considers CoAP-level features, thus improving on existing models by focusing on the application layer where CoAP operates. The use of Generative Adversarial Networks (GANs) to extend the CIDAD dataset from ~11,000 to 100,000 samples indicates a sophisticated approach to the data scarcity problem.
  • We thank the reviewer for the positive comments.

  1. However, the paper could benefit from a more in-depth discussion of the machine learning model itself. Details about the specific type of model used, its training process, and feature selection would be beneficial for understanding the solution more thoroughly and potentially replicating the study.

1) LinearSVC

Linear Support Vector Classifier (SVC) is a type of Support Vector Machine (SVM). SVM is one of the machine learning algorithms that is used for supervised learning (labeled data) such as detecting fraud, outliers and even regression problems. It simply draws a line between two categories by associating the similar data points in a separate class while the others in a different class. The result may contain different lines to classify the data points. Compared to K-Nearest Neighbor algorithm, SVC tries to classify the data point while mitigating the close data point to the line. This can be achieved by what-so-called decision boundary. So, SVC relies on the features to find the decision boundary and the line can be substituted by a hyperplane. However, in our work, LinearSVC performs worse with an accuracy of 59%. The confusion matrix and Receiver Operating Characteristic (ROC) show a lot of false positives and false negatives (Fig. 16 and Fig. 17). There are 7843 benign packets but mistakenly classified as malware. In addition, 4387 malware packets were wrongly classified as benign. It can be inferred that the false positives are higher than the false negatives using the LinearSVC algorithm. To avoid overfitting and underfitting, we validate the performance of the model using cross-validation technique with fold = 5. We apply the model on the training data and compare it to the performance of the model with testing data as shown in Table 4. The results show that the model is not suffering from overfitting or underfitting.

Table 4. Cross-Validation for LinearSVC

Fold No.

Fold 1

Fold 2

Fold 3

Fold 4

Fold 5

Model Accuracy

Performance

0.654

0.651

0.653

0.653

0.645

0.655

  1. Random Forest Classifier (RF) belongs to the decision tree algorithms family which rely on ensemble methods to avoid overfitting and underfitting that is common in traditional decision tree algorithms. The bagging methods are used to train RF by splitting the training data into sets, applying the decision tree for these sets and accumulate the results. Randomness and repetition of samples in RF is common, meaning a single instance may be used more than once due to recurrent sampling. In this work, the Random Forest Classifier shows better results than LinearSVC with an accuracy of 92%. However, the false positive rate is fair with total sample of 662 as shown in Fig. 18. Contrarily, the false negative is higher with 1651 samples. Fig. 19. Shows the ROC curve for the RF algorithm. To validate our results, we use cross-validation technique with fold = 5 as shown in Table 4.

Table 4. Cross-Validation for RF

Fold No.

Fold 1

Fold 2

Fold 3

Fold 4

Fold 5

Model Accuracy

Performance

0.924

0.924

0.925

0.923

0.923

0.92

3- Decision Tree (DT) is a famous algorithm in machine learning. It is suitable for classification and regression problems. The DT learning process is based on a sequence of comparisons between the data attributes (features) which result in more leaves that branch to the right or the left. When the learning reaches the end node, the decision is made based on the majority class in the leaf. A pre-learned threshold is set up to avoid infinite process for the learning. Compared to Neural Networks, DT performs well because it does not rely on gradient descent and the input normalization is not required. However, with image data, the neural networks outperform the DT. In our work, DT performs well with an accuracy of 98%. The number of false positives is 347 while the false negative rate is 302. The ROC curve infers the optimism of the decision tree algorithm. To validate the performance of DT, we apply cross-validation technique with fold = 5 as shown in Table 5.

Table 5. Cross-Validation for DT

Fold No.

Fold 1

Fold 2

Fold 3

Fold 4

Fold 5

Model Accuracy

Performance

0.985

0.985

0.985

0.984

0.987

0.987

4- Naïve Byes (NB) comes from the statistical methods based on Byes Theorem. It is a machine learning classifier that fit for classification problems. It is fast, accurate and perform well with large datasets. As the name imply, NB does not fetch for the relations between the features, assuming that each feature has independent impact to the decision. The learning process is based on calculating the prior probability of a class label, then the likelihood for each feature for each class is calculated. The result will be fed to the Byes formula to find the posterior probability. However, Naïve byes perform worse on our data and gain an accuracy of 70%. The false negative rate is higher than the false positives. The former is wrongly classified by the model with total samples of 6415 and 4612 for the latter. We validate the results using cross-validation technique as shown in Table 6.Fig. 22 and Fig. 23 show the confusion matrix and ROC curve for Naïve Byes performance.

Table 6. Cross-Validation for NB

Fold No.

Fold 1

Fold 2

Fold 3

Fold 4

Fold 5

Model Accuracy

Performance

0.702

0.706

0.707

0.707

0.710

0.70

This part is discussed in detail in the updated version.

  1. In the conclusion, the authors suggest future work involving the deployment of the model and its combination with other types of attacks for comprehensive testing. This provides a clear direction for subsequent research. Overall, the paper presents a promising solution to a pertinent issue in IoT security, though additional details about the machine learning model would enhance its impact and usability.

1) LinearSVC

Linear Support Vector Classifier (SVC) is a type of Support Vector Machine (SVM). SVM is one of the machine learning algorithms that is used for supervised learning (labeled data) such as detecting fraud, outliers and even regression problems. It simply draws a line between two categories by associating the similar data points in a separate class while the others in a different class. The result may contain different lines to classify the data points. Compared to K-Nearest Neighbor algorithm, SVC tries to classify the data point while mitigating the close data point to the line. This can be achieved by what-so-called decision boundary. So, SVC relies on the features to find the decision boundary and the line can be substituted by a hyperplane.

2) Random Forest Classifier (RF) belongs to the decision tree algorithms family which rely on ensemble methods to avoid overfitting and underfitting that is common in traditional decision tree algorithms. The bagging methods are used to train RF by splitting the training data into sets, applying the decision tree for these sets and accumulate the results. Randomness and repetition of samples in RF is common, meaning a single instance may be used more than once due to recurrent sampling.

3) Decision Tree (DT) is a famous algorithm in machine learning. It is suitable for classification and regression problems. The DT learning process is based on a sequence of comparisons between the data attributes (features) which result in more leaves that branch to the right or the left. When the learning reaches the end node, the decision is made based on the majority class in the leaf. A pre-learned threshold is set up to avoid infinite process for the learning. Compared to Neural Networks, DT performs well because it does not rely on gradient descent and the input normalization is not required. However, with image data, the neural networks outperform the DT.

4)  Naïve Byes (NB) comes from the statistical methods based on Byes Theorem. It is a machine learning classifier that fit for classification problems. It is fast, accurate and perform well with large datasets. As the name imply, NB does not fetch for the relations between the features, assuming that each feature has independent impact to the decision. The learning process is based on calculating the prior probability of a class label, then the likelihood for each feature for each class is calculated. The result will be fed to the Byes formula to find the posterior probability.

Elaborated in the updated version.

  1. Last but not least, the authors should elaborate more on the resource (hardware and time) consumption spent on the GANs models, and extensively comment on the complexity.

The CIDAD dataset is mainly targeting the CoAP with three different DDoS attacks: duplication, interception, and modification of the CoAP message with total malware samples of 288 only. Interception means intercepting stochastically sent packets before reaching the destination, whereas duplication is changing the content of the CoAP message, and modification is increasing the number of tokens.  The rest of the ~10,000 packets are benign packets. This poses an imbalance in the dataset since only 0.02% of the dataset is malware. So, we extend the dataset to 100,000 samples, of which ~50% for benign and ~50% are for malware. We used to use Generative Adversarial Network (GAN) to extend the 288 malware samples to ~50,000 samples. To do so, we use Google Colab platform [26] since the generated data needs high performance machine to manipulate the data. The time to generate the fake data from the malware is higher which take around one hour since we have only 288 samples and we aim to obtain ~50,000. However, the benign sample is adequate to generate the fake samples which take around 10 minutes to get the fake ones. The generated samples are compared with the original data to ensure the similarity between them by training each and calculating the Root Mean Square Error (RMSE). Our finding is surprising that the RMSE is ~0.005 for original data whereas the generated data is ~0.09. This indicates a coherent similarity between the original malware and the generated ones. On the other hand, for the benign samples, we do the same and find that RMSE for the original samples and the generated ones is ~0.03.

Also, the authors are suggested to recheck the plots well-scaled with enough resolution to illustrate.

Fixed in the updated version.

  • As per the reviewer’s comment, we have improved the language quality of the manuscript and thoroughly proofread for grammatical as well as typographical errors.

We thank the reviewer for the positive comments.

Reviewer 4 Report

The paper discusses the Internet of Things (IoT) and the security challenges associated with IoT devices. IoT is a network of interconnected devices that share information. However, the low power and resources of IoT devices make them vulnerable to attacks. The use of heavy protocols like HTTP is costly for IoT devices, leading to the implementation of lightweight protocols like CoAP. CoAP is popular but also targeted by attacks, including distributed denial of service (DDoS). Existing security models such as DTLS and LSPWSN have limitations. The paper proposes a machine learning model to detect DDoS attacks against CoAP with 98% accuracy. The CIDAD dataset is expanded using GANs, and the model outperforms existing models, achieving 93% accuracy in securing CoAP at the application layer.

Overall, the paper needs further improvement on several aspects. It looks like a report more than a research article.

1. The introduction should provide a more comprehensive explanation of the motivation behind this work, outlining the key reasons and significance of the research topic.

2. The paper should clearly identify the research gap and explicitly state the research questions that will be addressed. These research questions should guide the study and be clearly stated in the introduction or methodology section.

3. The literature review section requires further development. It should be expanded into a dedicated section that thoroughly discusses related works, existing frameworks, and provides a comparative analysis of different approaches based on various aspects such as performance, implementation, advantages, and limitations.

4. To enhance clarity and provide a concise overview of the contributions presented in recent studies, it would be beneficial to include tables summarizing and comparing the advantages, limitations, and other relevant aspects of these studies.

5. The quality of figures and tables should be improved to ensure legibility. Ensure that the text in these visuals is clear and not blurred, as this can impact the understanding of the presented information.

6. Figure 12 could be revised and presented in a more visually appealing and informative manner. Consider redesigning the figure to enhance its clarity and effectiveness in conveying the intended message.

7. It is important to discuss the limitations of the proposed method. Address the potential drawbacks or shortcomings of the approach and its implications. This discussion should precede the section on future work, allowing for a comprehensive understanding of the research outcomes and potential areas for improvement.

A major revision is required to improve the language. Many typos and grammtical issues have been found in the paper. The authors can use online platfomrs to improve the language.

Author Response

We would like to thank the reviewers and editors for providing an opportunity to revise the manuscript. We have studied these comments carefully and have made corresponding corrections that we hope will meet with your approval. In addition, as per the reviewer’s comment, we have improved the language quality of the manuscript and thoroughly proofread for grammatical as well as typographical errors. The revised text is mentioned in red font color.

The paper discusses the Internet of Things (IoT) and the security challenges associated with IoT devices. IoT is a network of interconnected devices that share information. However, the low power and resources of IoT devices make them vulnerable to attacks. The use of heavy protocols like HTTP is costly for IoT devices, leading to the implementation of lightweight protocols like CoAP. CoAP is popular but also targeted by attacks, including distributed denial of service (DDoS). Existing security models such as DTLS and LSPWSN have limitations. The paper proposes a machine learning model to detect DDoS attacks against CoAP with 98% accuracy. The CIDAD dataset is expanded using GANs, and the model outperforms existing models, achieving 93% accuracy in securing CoAP at the application layer.

Overall, the paper needs further improvement on several aspects. It looks like a report more than a research article.

  1. The introduction should provide a more comprehensive explanation of the motivation behind this work, outlining the key reasons and significance of the research topic.

Our motivation is based on the assumption that it is beneficial to secure the resource in its vicinity while keeping the mitigation near to the attack source. Existing models deploy DTLS to vet the CoAP message at the transport layer while CoAP operates over the application layer. Moreover, DTLS is vulnerable to DDoS attacks. So, in this work, we focus only on the CoAP-level features to fulfill the assumption by securing CoAP in its perimeter. The significance of this work comes from its novelty in detecting the DDoS packets at the application layer with high accuracy of 98%. Some of the existing methods pretend to collect features from all the layers of IoT network architecture, namely physical layer, network layer, transport layer, and application layer such as in reference 20. To sum up, securing the resource in its vicinity is recommended based on the researcher claimant as in reference 7.  

  1. The paper should clearly identify the research gap and explicitly state the research questions that will be addressed. These research questions should guide the study and be clearly stated in the introduction or methodology section.
  • The research gap is to find a method to secure CoAP in the application while not relying on the down layers to vet the CoAP message and deliver it to the application layer. The main aim of this research is to find a dataset containing different DDoS attacks and build a machine learning model that classifies these packets as a DDoS or benign in the application layer. The reach questions are as follows:

RQ1: Is it effective to secure CoAP in the application layer from DDoS attacks?

RQ2: What are the CoAP-level features that can be dedicated to secure CoAP in its perimeter?

RQ3: What is the best machine learning technique that can perform well in detecting DDoS attacks against CoAP?

The research gap and the research question are included in the updated version.

  1. The literature review section requires further development. It should be expanded into a dedicated section that thoroughly discusses related works, existing frameworks, and provides a comparative analysis of different approaches based on various aspects such as performance, implementation, advantages, and limitations.

In the literature section, we update it to “Related Technologies” and we remove the section that discusses DDoS against IoT network in general and focuses only on the defense mechanisms that dedicated for CoAP security only. So, the updated comparison between existing methods is shown in Table 1.

Table 1: IoT-CoAP defense mechanisms

Limitation

Results

Methodology used

Research objective

The assumption of third party (Proxy AC server) is trustworthy all the time.

Low computation time by delegating all handshake to a third party

DTLS handshake is extended with a cookie exchange technique to check the authentication of a message.

Detect and mitigate DDoS attack against CoAP(2016)

DTLS is computationally heavy for IoT devices.

Energy consumption, reduced packet size and reduced running time outperforms similar works

Trusted Third Party (TTP) is used to avoid DoS attack and reduce overhead on the server side.

Secure DTLS for IoT

(2017)

Anomaly-based detection may result in high false positive

60% efficiency when a honeypot is implemented

Deploy a honeypot with two phases, first to log the anomalies and second to verify or block the client

Deploy a honeypot to detect DDoS attack

(2017)

Focuses on packet payload feature only

Evaluate malicious packet delivery ratio and legitimate packet drop ratio

Compare the payload of benign and malicious packet, define a threshold and if exceeded, the source IP is blocked

Detect DDoS against IoT (2018)

Susceptible to spoofing attack

Fair energy and memory consumption when running the proposed system.

Relies on threshold by limiting the incoming request to a fixed number and if exceeded, source request is blocked.

Preventation framework from intrusion and DoS attack (2018)

High false positive rate of 20%

SVM classifies the anomalies with accuracy of 93%

Employ anomaly-based detection system to protect CoAP from DoS attack.

Protect 6LoWPAN and CoAP from DoS attack

(2018)

Some IoT devices have different regular patterns but not distinct patterns

RF, KNN, Neural Networks gain about 99% accuracy

Machine learning DDoS detection framework

Detect IoT DDoS attack (2018)

Susceptible to other kinds of attacks

Compared to D-WARD, FR-WARD performs better in retransmition, duration and energy consumption.

Leverage the fast retransmit and flow control mechanism of TCP to retransmit benign packets at fastest rate and malicious packet at harmless rate.

Defend IoT against DDoS while maintaining benign traffic (2018)

Susceptible to sniffing attack

Burp suite tool is used to intercept the communication between the client and the server

Set up a client/server architecture to check if the communication between the two devices using CoAP is secure.

Test CoAP MITM attack which results in spoofing, sniffing and DoS attack (2019)

N/A

Decrease overhead to the controller and CoAP responses become faster

SDN-based approach is developed to authorize the messages of CoAP

Securing CoAP messages (2020)

N/A

The proposed model outperforms the conventional system in terms of accuracy

 (IDS)-based deep learning integrated with blockchain to detect abnormal behavior in big networks.

Design blockchain enabled IDS with deep learning (2022)

N/A

Obtain accuracy of 99.05% with BoT-IoT dataset

Hybrid Harris Hawks with sine cosine and a deep learning-based intrusion detection system to detect DDoS attacks against IoT network

Blockchain-Assisted Hybrid Harris Hawks Optimization Based Deep DDoS Attack Detection in the IoT Environment (2023)

Due to limitation of sizes of the cells in the table, the comparison in terms of accuracy is included in “Proposed defense mechanisms for Securing CoAP against DDoS Attacks” as with supportive models architecture as follows:

2.2.1 Proposed defense mechanisms for Securing CoAP against DDoS Attacks

Some of the existing methods are proposed to secure the IoT network from DDoS attacks. Saveetha et al., (2022) claim that the intruder needs to discover the mapping of a network and it is hard to track all the scanning processes due to huge networks implementations. Consequently, the author developed an intrusion detection system (IDS) integrated with blockchain to detect the intrusions. Katib et al., (2023) claim that Blockchain has a significant role in IoT-based applications. Blockchain is used in many aspects such as security and privacy in IoT-enabled deployment. The authors proposed a hybrid Harris Hawks with sine cosine and a deep learning-based intrusion detection system to detect DDoS attacks against IoT network. The BoT-IoT dataset was used to test their method and the model shows impressive accuracy of 99.05%. However, these works are comprehensively dedicated for detecting attacks against IoT network while our focus is securing CoAP specifically from DDoS attacks since DDoS can target any layer on the IoT network architecture.

2.2.1.1 DTLS for CoAP Security

Some researchers proposed that DTLS be implemented in CoAP for security purposes. Maleh et al., (2016) mentioned that Datagram Transport Layer (DTLS) handshake suffers from spoofing IP address DDoS attacks [15]. To face this issue, the DTLS handshake is expanded with a cookie exchange technique. The capability and threshold for receiving packets must be declared with the IP address to the server, then, the server reserves resources for new communication. Because of the costly energy of this technique, they moved it to Proxy AC Server with no energy constraints. Their method is depicted in Fig. 9, and they claim it reduces the resource occupancy DTLS ROM by 23% compared to standard DTLS. Haroon A. et al. (2017) proposed an enhancement to DTLS to make it resistant to DDoS attacks [16]. They claim that their method can reduce the overhead of handshaking time, packet size, and energy consumption compared to other works. The authors’ system named E-lithe relies on Trusted Third Party (TTP) to reduce the overhead on the server-side. Compared to Lithe and other works, E-lithe outperforms others in terms of running time and reduced packet size. Later, this work was enhanced by Shruti et al., (2018) who claimed that their work outperforms E-Lithe [17] as depicted in Fig. 10. The authors’ work essentially focuses on the comparison between the payload of benign and malicious packets, defining a threshold, if the threshold is exceeded, then the source IP is blocked. They evaluate their work based on the malicious packet delivery ratio and legitimate packet drop ratio which outperforms the work done by Haroon et al.

Fig. 9. Cookie Exchange Technique by Maleh el al. [15]

Fig 10. Comparison between E-Lithe and Shruti et al. work [17]

2.2.1.2 SDN for CoAP Security

Alzahrani et al. (2020) implemented a software-defined networking scheme (depicted in Fig. 11) to authorize the messages over the CoAP protocol [18]. They argue that distributed approach in which IoT devices employ powerful gateways attached to them may be insufficient and making the access control decisions accomplished by the controller renders the security of CoAP messages more efficient and can help to avoid DDoS attacks.

Fig 11. SDN-based schema for CoAP message authorization [18]

2.2.1.3 Machine Learning for CoAP Security

Machine Learning is also proposed to detect and mitigate DDoS attacks against CoAP. Granjal et al., (2018) developed a framework that employs a threshold to mitigate DDoS attacks [19]. They define a limit for messages of CoAP, and after the limit is exceeded, extra messages will be dropped. The authors enhanced their work and proposed an anomaly-based detection system to protect the 6LoWPAN and CoAP protocol from DDoS attacks [20] depicted in Fig. 12. SVM is used as an ML-Classifier and gains an accuracy of 93%. However, the system generates a high false-positive rate of around 20%. Doshi et al., (2018) developed a machine learning pipeline (depicted in Fig. 13) that is employed on middleboxes such as routers or firewalls to detect IoT DDoS attacks [20]. They claim that IoT traffic is distinct from other traffics coming from other internet-connected devices because IoT traffic is repetitive and often communicates with a small finite of endpoints. After testing this method, it gains an accuracy of 99% using RF, KNN, and Neural Networks.

Fig. 12. Anomaly-based detection framework for CoAP DDoS attacks by Granjal et al. [18]

2.2.1.4 Other Methods for CoAP Security

Other works propose different methods to cope with DDoS attacks against CoAP. Anirudh et al. (2017) propose a honeypot to lure the attacker and log his information for future verification or block purposes.

Fig 13. Machine Learning-based pipeline to mitigate CoAP DDoS attacks by Doshi et al. [20]

Fig 14. Honypot-based detection method for mitigating CoAP DDoS attacks by Anirudh et al. [21]

  1. To enhance clarity and provide a concise overview of the contributions presented in recent studies, it would be beneficial to include tables summarizing and comparing the advantages, limitations, and other relevant aspects of these studies.

Table 1: IoT-CoAP defense mechanisms

Limitation

Results

Methodology used

Research objective

The assumption of third party (Proxy AC server) is trustworthy all the time.

Low computation time by delegating all handshake to a third party

DTLS handshake is extended with a cookie exchange technique to check the authentication of a message.

Detect and mitigate DDoS attack against CoAP(2016)

DTLS is computationally heavy for IoT devices.

Energy consumption, reduced packet size and reduced running time outperforms similar works

Trusted Third Party (TTP) is used to avoid DoS attack and reduce overhead on the server side.

Secure DTLS for IoT

(2017)

Anomaly-based detection may result in high false positive

60% efficiency when a honeypot is implemented

Deploy a honeypot with two phases, first to log the anomalies and second to verify or block the client

Deploy a honeypot to detect DDoS attack

(2017)

Focuses on packet payload feature only

Evaluate malicious packet delivery ratio and legitimate packet drop ratio

Compare the payload of benign and malicious packet, define a threshold and if exceeded, the source IP is blocked

Detect DDoS against IoT (2018)

Susceptible to spoofing attack

Fair energy and memory consumption when running the proposed system.

Relies on threshold by limiting the incoming request to a fixed number and if exceeded, source request is blocked.

Preventation framework from intrusion and DoS attack (2018)

High false positive rate of 20%

SVM classifies the anomalies with accuracy of 93%

Employ anomaly-based detection system to protect CoAP from DoS attack.

Protect 6LoWPAN and CoAP from DoS attack

(2018)

Some IoT devices have different regular patterns but not distinct patterns

RF, KNN, Neural Networks gain about 99% accuracy

Machine learning DDoS detection framework

Detect IoT DDoS attack (2018)

Susceptible to other kinds of attacks

Compared to D-WARD, FR-WARD performs better in retransmition, duration and energy consumption.

Leverage the fast retransmit and flow control mechanism of TCP to retransmit benign packets at fastest rate and malicious packet at harmless rate.

Defend IoT against DDoS while maintaining benign traffic (2018)

Susceptible to sniffing attack

Burp suite tool is used to intercept the communication between the client and the server

Set up a client/server architecture to check if the communication between the two devices using CoAP is secure.

Test CoAP MITM attack which results in spoofing, sniffing and DoS attack (2019)

N/A

Decrease overhead to the controller and CoAP responses become faster

SDN-based approach is developed to authorize the messages of CoAP

Securing CoAP messages (2020)

  • Enhanced in the updated version.
  1. The quality of figures and tables should be improved to ensure legibility. Ensure that the text in these visuals is clear and not blurred, as this can impact the understanding of the presented information.

                                        Fig. 9. Cookie Exchange Technique by Maleh el al. [15]

Fig 10. Comparison between E-Lithe and Shruti et al. work [17]

Fig 11. SDN-based schema for CoAP message authorization [18]

Fig. 12. Anomaly-based detection framework for CoAP DDoS attacks by Granjal et al. [18]

Fig 13. Machine Learning-based pipeline to mitigate CoAP DDoS attacks by Doshi et al. [20]

Fig 14. Honypot-based detection method for mitigating CoAP DDoS attacks by Anirudh et al. [21]

Fixed in the updated version.

  1. Figure 12 could be revised and presented in a more visually appealing and informative manner. Consider redesigning the figure to enhance its clarity and effectiveness in conveying the intended message.

Figure 12 visualizes the distribution of the dataset and represents the percentage of benign samples and the malware samples in the dataset. Both classes have ~50% samples for each to ensure the dataset is balanced. The Figure is updated as follows.

Fig. 12. Dataset distribution

Fixed in the updated version.

  1. It is important to discuss the limitations of the proposed method. Address the potential drawbacks or shortcomings of the approach and its implications. This discussion should precede the section on future work, allowing for a comprehensive understanding of the research outcomes and potential areas for improvement.

In the conclusion section, we conclude that implementing the proposed method in a real-world scenario is recommended to test the performance. In addition, combining the dataset with other attacks such as DDoS amplification attacks and enumeration attacks will result in a coherent dataset for CoAP DDoS attacks. So, these future works represent the limitations in our work.

Fixed in the updated version.

  1. A major revision is required to improve the language. Many typos and grammatical issues have been found in the paper. The authors can use online platforms to improve the language.
  • As per the reviewer’s comment, we have improved the language quality of the manuscript and thoroughly proofread for grammatical as well as typographical errors.

We thank the reviewer for the positive comments.

Round 2

Reviewer 2 Report

The work entitled “Application Layer-Based Denial-of-Service Attacks Detection Against IoT-CoAP” by Sultan Almeghlef and collaborators describes the generation of a dataset using Generative Adversarial Network (GAN) to train a model to identify benign IoT communications from malware.

All the concerns were properly addressed.

It is advisable for authors to include their GAN data in a public repository, such as GitHub or others, so this data may be used/validated/improved by others.

Author Response

We would like to thank the reviewers and editors for providing an opportunity to revise the manuscript. We have studied these comments carefully and have made corresponding corrections that we hope will meet with your approval. In addition, as per the reviewer’s comment, we have improved the language quality of the manuscript and thoroughly proofread for grammatical as well as typographical errors. The revised text is mentioned in red font color.

The work entitled “Application Layer-Based Denial-of-Service Attacks Detection Against IoT-CoAP” by 
Sultan Almeghlef and collaborators describes the generation of a dataset using Generative Adversarial 
Network (GAN) to train a model to identify benign IoT communications from malware.

1) All the concerns were properly addressed.
Thank you for the positive comment.

2) It is advisable for authors to include their GAN data in a public repository, such as GitHub or others, so this data may be used/validated/improved by others.
Based on the reviewer comment, we have Uploaded GAN data to Kaggle a week ago.
https://www.kaggle.com/datasets/salmeghlef/ddos-coap-dataset-cidad
Thank you for the suggestion.

We thank the reviewer for the valuable comments.

Reviewer 4 Report

The authors have addressed all my comments, I have no further suggestions.

The english has been improved compared to the initial version.

Author Response

We would like to thank the reviewers and editors for providing an opportunity to revise the manuscript. We have studied these comments carefully and have made corresponding corrections that we hope will meet with your approval. In addition, as per the reviewer’s comment, we have improved the language quality of the manuscript and thoroughly proofread for grammatical as well as typographical errors. The revised text is mentioned in red font color.

  • The authors have addressed all my comments, I have no further suggestions.

Thank you for the positive comment.

  • Comments on the Quality of English Language

The English has been improved compared to the initial version.

Based on the reviewer’s comment, we have comprehensively improved the language quality of the manuscript and thoroughly proofread for grammatical as well as typographical errors.

We thank the reviewer for the valuable comments.
